# Rapid adaptation of a complex trait during experimental evolution of *Mycobacterium tuberculosis*

Tracy M Smith[1†], Madison A Youngblom[2,3†], John F Kernien[2], Mohamed A Mohamed[2], Sydney S Fry[2], Lindsey L Bohr[2,3], Tatum D Mortimer[4], Mary B O'Neill[5], Caitlin S Pepperell[2,6]*

[1]New York Genome Center, New York, United States; [2]Department of Medical Microbiology and Immunology, School of Medicine and Public Health, University of Wisconsin-Madison, Madison, United States; [3]Microbiology Doctoral Training Program, University of Wisconsin-Madison, Madison, United States; [4]Department of Immunology and Infectious Diseases, Harvard T.H. Chan School of Public Health, Boston, United States; [5]Laboratoire de Biochimie (LBC), Chimie Biologie et Innovation, ESPCI Paris, PSL Université, Paris, France; [6]Department of Medicine (Infectious Diseases), School of Medicine and Public Health, University of Wisconsin-Madison, Madison, United States

**\*For correspondence:**
cspepper@medicine.wisc.edu

†These authors contributed equally to this work

**Abstract** Tuberculosis (TB), caused by *Mycobacterium tuberculosis* (*M. tb*), is a leading cause of death due to infectious disease. TB is not traditionally associated with biofilms, but *M. tb* biofilms are linked with drug and immune tolerance and there is increasing recognition of their contribution to the recalcitrance of TB infections. Here, we used *M. tb* experimental evolution to investigate this complex phenotype and identify candidate loci controlling biofilm formation. We identified novel candidate loci, adding to our understanding of the genetic architecture underlying *M. tb* biofilm development. Under selective pressure to grow as a biofilm, regulatory mutations rapidly swept to fixation and were associated with changes in multiple traits, including extracellular matrix production, cell size, and growth rate. Genetic and phenotypic paths to enhanced biofilm growth varied according to the genetic background of the parent strain, suggesting that epistatic interactions are important in *M. tb* adaptation to changing environments.

## Editor's evaluation

This study is of relevance to microbiologists interested in understanding the mechanism of biofilm development and notably in *Mycobacterium tuberculosis* (Mtb). The authors describe the use of experimental evolution for identifying genes/loci involved in controlling biofilm formation and report a few fixed mutations/genetic duplications that could be associated with biofilm formation or its regulation. The findings also reveal epistatic interactions among genes that are consequential during growth of Mtb in biofilms.

## Introduction

In 2019, an estimated 10 million people fell ill due to tuberculosis (TB), and one quarter of the world's population is estimated to be infected with its causative agent *Mycobacterium tuberculosis* (*M. tb*) (***Global Tuberculosis Report, 2019***, WHO). New strategies for diagnosis, treatment, and control of TB are urgently needed. From an evolutionary perspective, *M. tb* stands out among bacterial

**eLife digest** In many environments, bacteria live together in structures called biofilms. Cells in biofilms coordinate with each other to protect the group and allow it to survive difficult conditions. *Mycobacterium tuberculosis*, the bacterium that causes tuberculosis, forms biofilms when it infects the human body. Biofilms make the infection a lot more difficult to treat, which may be one of the reasons why tuberculosis is the deadliest bacterial infection in the world.

Bacteria evolve rapidly over the course of a single infection, but bacteria forming biofilms evolve differently to bacteria living alone. This evolution happens through mutations to the bacterial DNA, which can be small (a single base in a DNA sequence changes to a different base) or larger changes (such as the deletion or insertion of several bases).

Smith, Youngblom et al. studied the evolution of tuberculosis growing in biofilms in the lab. As the bacteria evolved, they tended to form thicker biofilms, an effect linked to 14 mutations involving single base DNA changes and four larger ones. Most of the changes were in regulatory regions of DNA, which control whether genes are 'read' by cells to produce proteins. These regions often change more though evolution than regions coding for proteins, because they have a coordinated effect on a group of related genes rather than randomly altering individual genes. Smith, Youngblom et al. also showed that biofilms made from different strains of tuberculosis evolved in different ways.

Smith Youngblom et al.'s findings provide more information regarding how bacteria adapt to living in biofilms, which may reveal new ways to control them. This could have applications in water treatment, food production and healthcare. Learning how to treat bacteria growing in biofilms could also improve the outcomes for patients infected with tuberculosis.

pathogens for its strict association with human hosts, limited genetic diversity, and clonal evolution (*Eldholm and Balloux, 2016*). We might expect these features to constrain adaptation of *M. tb*, yet TB remains a challenging infection to treat due to the bacterium's ability to persist in the face of antibiotic and immune pressure and to acquire novel drug resistances. In order to better treat and control TB, we need to understand the sources of *M. tb*'s robustness and to identify its vulnerabilities. Experimental evolution is a powerful tool for illuminating these strengths and vulnerabilities and has led to important insights into the fundamental processes guiding microbial adaptation.

Biofilms are increasingly recognized as a relevant growth form for bacteria in their natural environments (*Costerton et al., 1999*). TB is not traditionally thought of as a biofilm infection. However, *M. tb* cells spontaneously aggregate and secrete extracellular matrix (ECM) when grown in vitro (*Dubos and Davis, 1946*; *Bacon et al., 2014*), suggesting they are naturally inclined to grow as biofilms. Autopsy studies have long identified *M. tb* aggregates in human tissues during TB infection (*Canetti, 1956*; *Nyka, 1977*; *Nyka, 1967*; *Nyka, 1963*; *Nyka and O'Neill, 1970*). More recent research has demonstrated specific biomarkers of *M. tb* biofilms in human autopsy specimens and animal models of TB (*Chakraborty et al., 2021*). The presence of *M. tb* biofilms during TB infection is of major practical significance as growth within a biofilm allows *M. tb* cells to survive otherwise lethal concentrations of antibiotics and to evade immune responses (*Ojha et al., 2008*; *Ackart et al., 2014*; *Trivedi et al., 2016*; *Chakraborty et al., 2021*). Identifying the mechanisms of biofilm development by *M. tb* can thus aid the development of new, more effective therapies for TB (*Wang et al., 2013*; *Ackart et al., 2014*; *Richards et al., 2019*; *Chakraborty et al., 2021*).

The genetic determinants of *M. tb* biofilm formation have been investigated with candidate gene approaches and phenotypic characterization of knockout, knockdown, and overexpression mutants (*Ojha et al., 2008*; *Pang et al., 2012*; *Sambandan et al., 2013*; *Wolff et al., 2015*; *Rastogi et al., 2017*; *Yang et al., 2017*; *Richards et al., 2019*; *Hegde, 2019*; *Bharti et al., 2021*; *Chakraborty et al., 2021*). Here, we use a complementary approach based on serial passaging of *M. tb* clinical isolates under selective pressure to grow as a biofilm. This approach has the advantage of maintaining the integrity of complex networks of genes and their regulators while enabling discovery of subtle genetic changes with an impact on biofilm phenotype. It is also unbiased with respect to the choice of candidate loci. During experimental evolution of six closely related *M. tb* isolates passaged over months to years we found: (1) rapid adaptation in response to selection imposed in our system, with development of more robust biofilm growth in all strains, (2) changes in a range of *M. tb* phenotypes

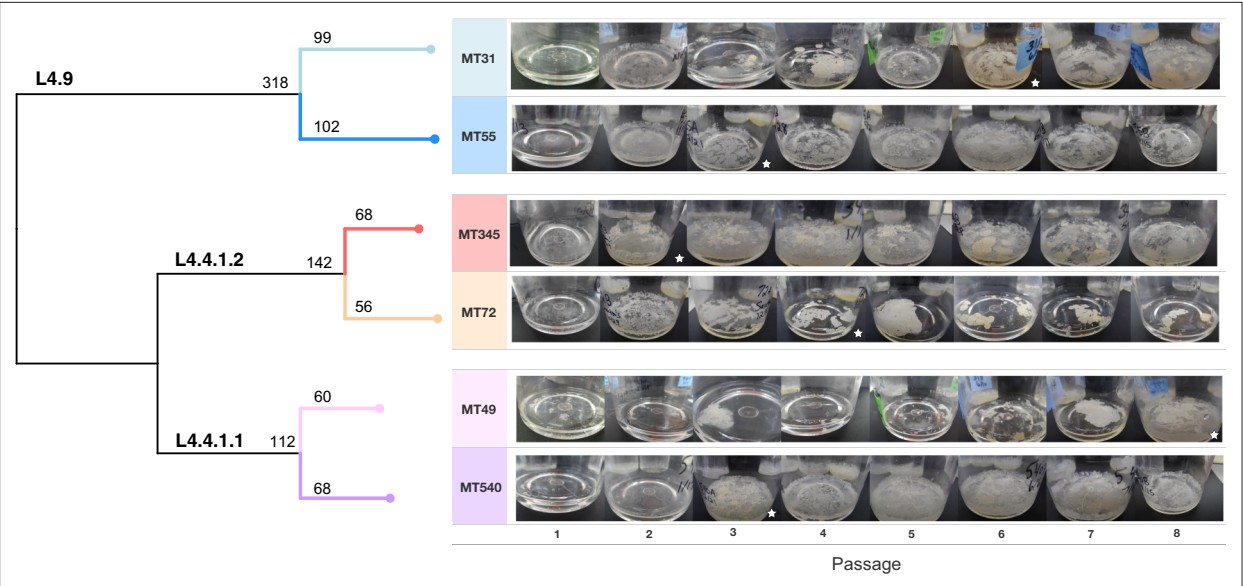

**Figure 1.** Whole genome sequence phylogeny of ancestral populations used for passaging and photos of pellicles throughout the experiment. Populations fall into three clades with single-nucleotide polymorphism distances given for each sub-lineage and strain. Pellicle photos show rapid change in phenotype, followed by stabilization through the rest of the experiment. The star indicates the passage at which point we determined the pellicle phenotype to be stabilized for each population. Photos of pellicles were taken after 5–7 weeks of growth.

The online version of this article includes the following figure supplement(s) for figure 1:

**Figure supplement 1.** All photos from pellicle passaging experiment.

in association with few, presumably pleiotropic mutations, (3) predominant impacts on gene dosage among mutations that emerged during the experiment, (4) implication of *M. tb* loci not previously known to be involved in biofilm development, and (5) apparent effects of strain genetic background in shaping the adaptive path to the phenotype under selection.

## Results
### Sample

Pellicles are a specific type of biofilm in which bacteria form aggregates at air–liquid interfaces (*Kobayashi, 2007*); *M. tb* has been shown to form pellicle biofilms in vitro (*Kerns et al., 2014*). For our study, we chose six closely related isolates of *M. tb*, from three sub-clades of the Euro-American lineage L4 (4.9, 4.4.1.2, and 4.4.1.1 shown in *Figure 1*). The study design enabled us to identify the impacts of genetic background over multiple scales, including comparisons between sub-lineages (4.9 and 4.4), sub-sub-lineages (4.4.1.2 and 4.4.1.1), and individual strains. In addition, the strain selection encompassed a variety of biofilm phenotypes such that we could identify impacts of ancestral phenotype on bacterial adaptation in our passaging system. The strains were grown as pellicles following a published protocol (*Kulka et al., 2012*). We passaged these six populations in pellicle form every 5–7 weeks as described in the Methods section. Each pellicle population was passaged at least eight times over a period of 2 years (*Figure 1—figure supplement 1*; *Supplementary file 1*). To investigate the specificity of adaptations observed during pellicle passaging, we also passaged the six strains in planktonic culture: we passaged three independently evolving populations per strain, and each population was passaged four times.

### Phenotypic changes of pellicles

At each passage, we photographed the pellicle and described its growth according to the following criteria: proportion of liquid surface covered, presence of climbing (attachment to and growth up the sides of the flask), thickness of growth, and continuity of growth (versus discontinuous patches). Although the *M. tb* strains were closely related (i.e. separated by 100 s of single-nucleotide polymorphisms [SNPs]), differences in biofilm phenotype were evident prior to passaging (*Figure 1*). During

the initial few passages, phenotypes changed for all strains and then stabilized between passages 2 and 8 (*Figure 1*). We performed extended passaging for four strains, which were carried out to passage 16 or 20; we did not observe any further phenotypic changes during this extended passaging (*Figure 1—figure supplement 1*).

Over the course of the experiment, all populations evolved more robust biofilms characterized by an increase in surface coverage and thicker, reticulating growth (*Figure 1*). MT72 was an exception, as it developed a typical confluent biofilm after two passages, but then evolved discontinuous, thick growth covering only a small fraction of the surface. This contrasts with the other evolved pellicles, which cover the entire surface of the liquid, climb up the side of the flask, and are more confluent (*Figure 1*).

To further characterize *M. tb* biofilm phenotypes, we performed SEM of biofilm samples from ancestral strains and paired evolved strains that had undergone eight rounds of passaging. The appearance of ECM was variable among strains at baseline, but all strains exhibited increased production of ECM after serial passaging (*Figure 2*). We observed changes in ECM appearance including increased webbing (MT55 and MT540) and increased production of globules (MT31, MT345, MT72, and MT49) (*Figure 2*). Bacterial cell shape has been documented to have direct fitness consequences in many species (*Yang et al., 2016*). We measured bacilli length of ancestral and evolved strains and found that three strains (MT55, MT72, and MT345) had significant changes in cell length (*Figure 2*). The direction of the change varied among strains: MT55 and MT72 evolved longer bacilli whereas MT345 evolved shorter bacilli (*Figure 2*).

## Wet weights

To obtain another quantitative measure of phenotypic adaptation to biofilm growth, we developed a protocol for measuring the wet weight of a pellicle biofilm. Wet weights of ancestral biofilms were similar except for MT345, which was heavier in keeping with its confluent morphotype (*Figures 1 and 3*). We observed a significant (Mann-Whitney U test, p-value=1.5e-06) increase in wet weight for all populations after eight passages (*Figure 3*). Of note, we observed a substantial increase in wet weight in MT345, which formed a robust pellicle at baseline, suggesting that the potential to increase pellicle biomass remains among *M. tb* isolates that form robust biofilms at baseline. Indeed, MT345 along with MT55 had particularly dramatic increases in pellicle biomass (an average of 2.3 g and 3.5 g, respectively) with evolved isolates approaching the phenotype of lab-adapted strain H37Rv (*Figure 3*). We also observed a trend towards increased variability of wet weights between replicate cultures of the evolved biofilms (*Figure 3*).

## Planktonic growth trade-offs during pellicle adaptation

To investigate whether adaptation to pellicle growth involved trade-offs with fitness in planktonic culture, we compared planktonic growth rates of ancestral and pellicle-evolved populations. Some evolved strains grew faster in planktonic culture following pellicle passaging (MT31, MT345, and MT49) whereas other strains exhibited similar growth rates – if slightly different kinetics – when compared with their ancestral strains (MT55, and MT72; *Figure 4*). One evolved strain (MT540) had a slower growth rate than its ancestor (*Figure 4C*; E).

## Candidate loci involved in biofilm formation

Genomic DNA (gDNA) was extracted from whole biofilm populations for (pooled) sequencing every four passages (except MT31 and MT49 which were first sequenced at passage 8; *Supplementary file 1*). To identify loci potentially responsible for the observed changes in pellicle phenotype, we identified mutations that rose to fixation (variant at 100% in the population) or disappeared (variant at 0% and reference allele at 100%) over the course of our experiment. After filtering and manual curation of the results (see Methods), we identified 14 SNPs, 2 duplications, and 2 deletions that met these criteria (*Figure 5*). The starting frequency of these alleles varied from 0 to 55% and fixation occurred by the eighth passage for all but two of these variants (position 4,115,505 in MT31 and position 1,416,232 in MT72; *Figure 5*). Candidate genes are annotated with a range of functions with regulation (*regX3, phoP, embR,* and *Rv2488c*) and lipid metabolism (*Rv2186c, mgtA,* and *fadE22*) being the most common (*Figure 5*). A search for these mutations within a global dataset containing ~40,000 isolates (see Methods) revealed these mutations to be exceedingly rare or completely absent from other

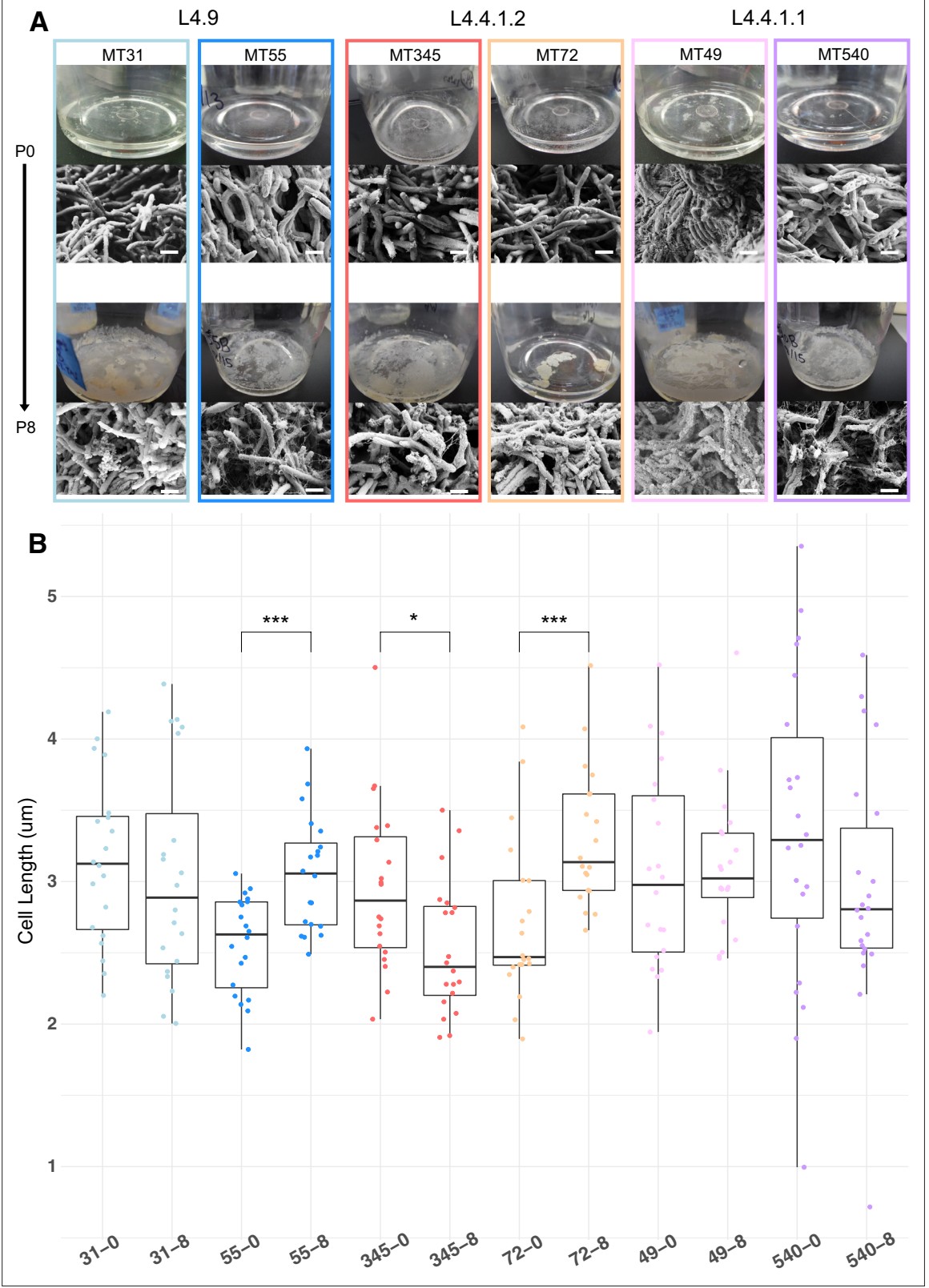

**Figure 2.** Phenotypic changes in evolved bacterial populations: cell size & matrix production. (**A**) SEM images of ancestral and evolved biofilms show changes in extracellular matrix and (**B**) cell length. SEMs were taken after 5 weeks of biofilm growth, shown alongside photo of pellicle from that passage. Cell lengths were measured across two biological replicates for each strain, except MT540-0 for which only one replicate was available. Each dot represents a single cell length measurement. Significant differences (Kruskal–Wallis p-value=3.43e-05) in cell length between ancestral and evolved

*Figure 2 continued on next page*

*Figure 2 continued*

pairs shown across the top of panel B (Mann-Whitney U test with Benjamini-Hochberg correction). Scale bars in lower right side of SEM images = 1 μm. p-value legend: *p<0.05, **p<0.01, ***p<0.001.

The online version of this article includes the following source data for figure 2:

**Source data 1.** Cell lengths calculated by SEM for ancestral and pellicle evolved strains shown in *Figure 2B*.

sequenced strains of *M. tb* (*Figure 5*). The most commonly identified SNP was at position 3,690,947 upstream of *lpdA* (identified in our study in strain MT49), at a frequency of 0.0015 in this dataset (*Figure 5*). One caveat is that genes in repetitive families (PE, PPE, and PE-PGRS) were excluded from our analysis due to difficulties in accurately resolving these regions with short-read sequencing data. It is possible that variants contributing to enhanced biofilm formation lie undetected in these regions.

We did not identify any overlap in mutations that arose following pellicle and planktonic passaging, either in specific SNPs or genes containing SNPs (*Figure 5*, *Supplementary file 2*). This suggests that the candidate variants for pellicle adaptation are unlikely to represent generalized lab adaptations.

Allele frequency dynamics for all variants that changed >30% over the course of pellicle passaging are shown in *Figure 6*. In most populations (MT31, MT345, MT72, and MT540), we observed secondary mutations arising on the background of the first fixed mutations (*Figure 6*). Some secondary mutations reached high frequencies (>95%) but did not remain at high frequency for the duration of the experiment (*Figure 6B*). We identified a single mutation in the evolved pellicles (*Figure 6A*, P286L *Rv2319c*) with a moderate change in frequency (>30%) that did not become fixed. Allele frequency trajectories were otherwise characterized by rare variants that remained rare. Few changes in allele frequency were observed for MT55. Aside from the large duplication that reached fixation, we did not identify any other mutations in this strain with ≥30% change in frequency.

We calculated genome-wide measures of diversity (Tajima's D and nucleotide diversity pi) for sequence data at each timepoint (*Figure 6—figure supplement 1*). Our values concur with previously estimated values for *M. tb* (*O'Neill et al., 2015*) and are consistent with genome-wide purifying selection (*Pepperell et al., 2010*; *Pepperell et al., 2013*). Bacterial populations within pellicles were more diverse (characterized by higher values of pi) and exhibited a stronger skew to rare variants (lower Tajima's D) than those found in planktonically grown communities (*Figure 6—figure supplement 1*). This observation points to the complexity of biofilm communities and distinct demographic and/or selective forces associated with pellicle versus planktonic growth.

## Convergent adaptation

We observed two instances of convergent adaptation, i.e., the same locus being subject to repeated mutation during pellicle adaptation. Interestingly, each pair of convergent mutations appeared in the same sub-lineage. A large (~120 kb) duplication fixed in populations MT31 and MT55, which both belong to sub-lineage L4.9. The second convergent mutation, an intergenic SNP upstream of NAD(P) H quinone reductase *lpdA*, within a transcription factor binding site (TFBS) and a non-coding RNA (ncRNA), occurred in MT49 and MT540 that both belong to sub-lineage L4.4.1.1 (*Figure 5*). Similar patterns were evident in the planktonically passaged populations of bacteria: we identified convergence in these experiments, with mutations arising repeatedly at the same loci (*Supplementary file 2*, *Figure 6—figure supplement 2*). We also observed a likely impact of genetic background on adaptation, with loci subject to repeated mutation in an apparently strain- or sub-lineage specific manner (*Supplementary file 2*, *Figure 6—figure supplement 2*).

## Convergent adaptation affecting expression of *lpdA*

We identified convergent adaptation in evolved pellicles of strains MT49 and MT540 (both of lineage L4.4.1.1) in the region upstream of NAD(P)H quinone reductase *lpdA* (*Figure 5*). These intergenic SNPs are five base pairs apart and lie within the TFBS for *Rv1719*, which is known to downregulate *lpdA* (*Rustad et al., 2014*; *Figure 7A*). We hypothesized that an SNP within the TFBS would affect transcription factor (TF) binding and therefore expression of *lpdA*. We sought to quantify *lpdA* expression by quantitative polymerase chain reaction (qPCR) in ancestral populations and compare that to expression in evolved populations containing the SNP of interest. Evolved populations had significantly higher (p<0.05, Mann-Whitney U test) expression of *lpdA* (*Figure 7*). Additionally, we

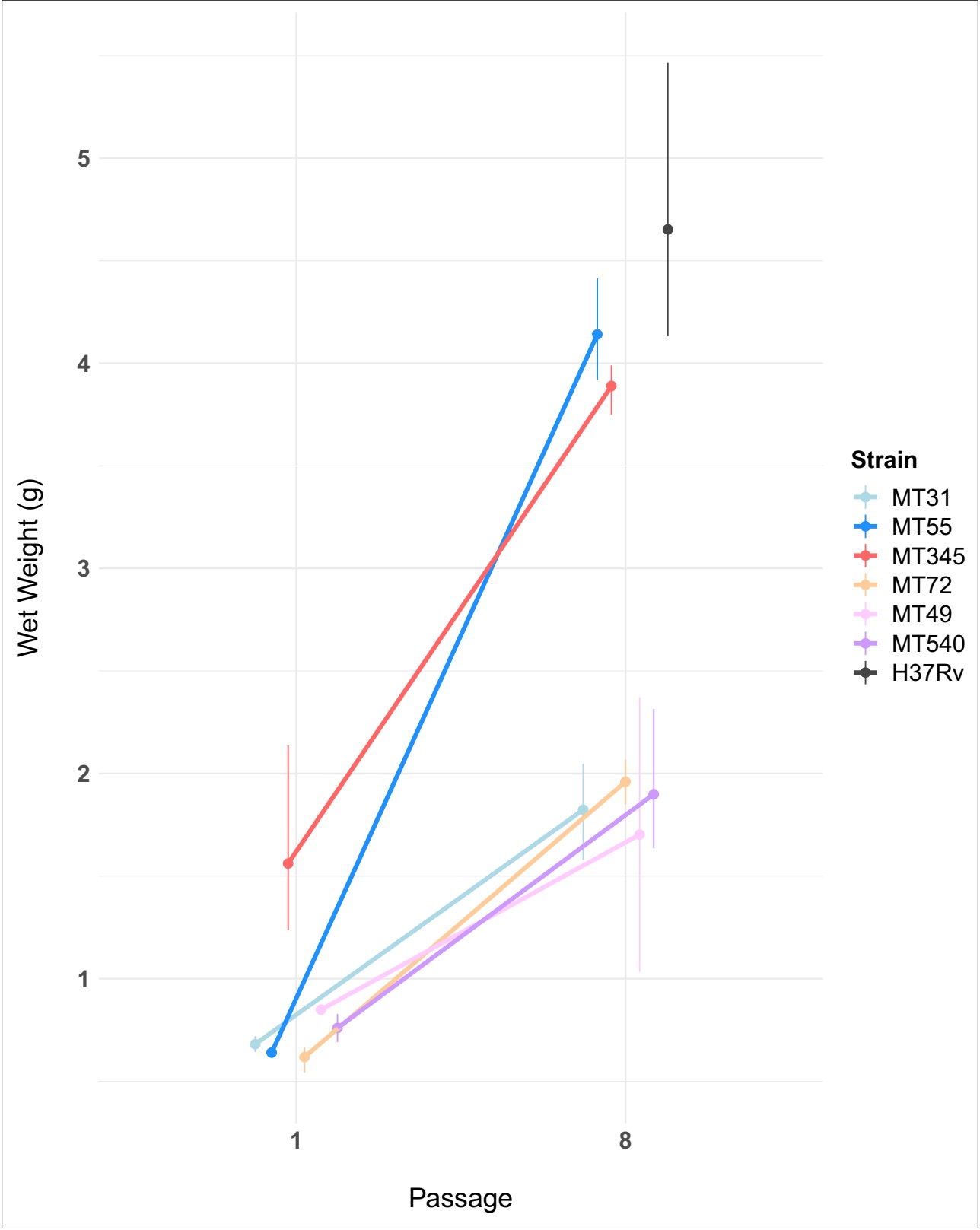

**Figure 3.** Pellicle wet weights measured after 5 weeks of growth, for ancestral populations and populations after eight passages (n=2–3 per experimental strain, n=5 for H37Rv, see *Figure 3—source data 1*). Pellicle wet weights increased for all experimental populations following passaging (Mann-Whitney U test, p-value=1.5e-06). The magnitude of this change varied, with two genetic backgrounds (MT55 & MT345) showing relatively

*Figure 3 continued on next page*

*Figure 3 continued*
dramatic increases in pellicle weight, approaching the phenotype of lab-adapted strain H37Rv. We also observed a trend towards increased variability in wet weights after passaging as a pellicle. Error bars represent range of wet weights across replicates.

The online version of this article includes the following source data for figure 3:

**Source data 1.** Pellicle wet weights of ancestral and pellicle evolved strains.

observed significantly increased expression of *lpdA* during biofilm growth when compared with planktonic growth, suggesting a role for this gene in biofilm formation (*Figure 7*). In addition to being within a TFBS affecting expression of *lpdA*, these SNPs also lie within an ncRNA that is expressed from the opposite strand of *lpdA* (*Figure 7*). This ncRNA was identified by RNA-sequencing (*Arnvig et al., 2011*; *Gerrick et al., 2018*) but has yet to be characterized. In order to further investigate the role of *lpdA* in pellicle growth, we measured pellicle phenotypes in H37Rv after introducing a second copy of the gene with an integrative, constitutive overexpression plasmid. Pellicles from bacteria with the additional copy of *lpdA* had significantly higher mass than those from bacteria transformed with an empty vector (*Figure 7C*), with no change in gross phenotype (*Figure 6—figure supplement 1*).

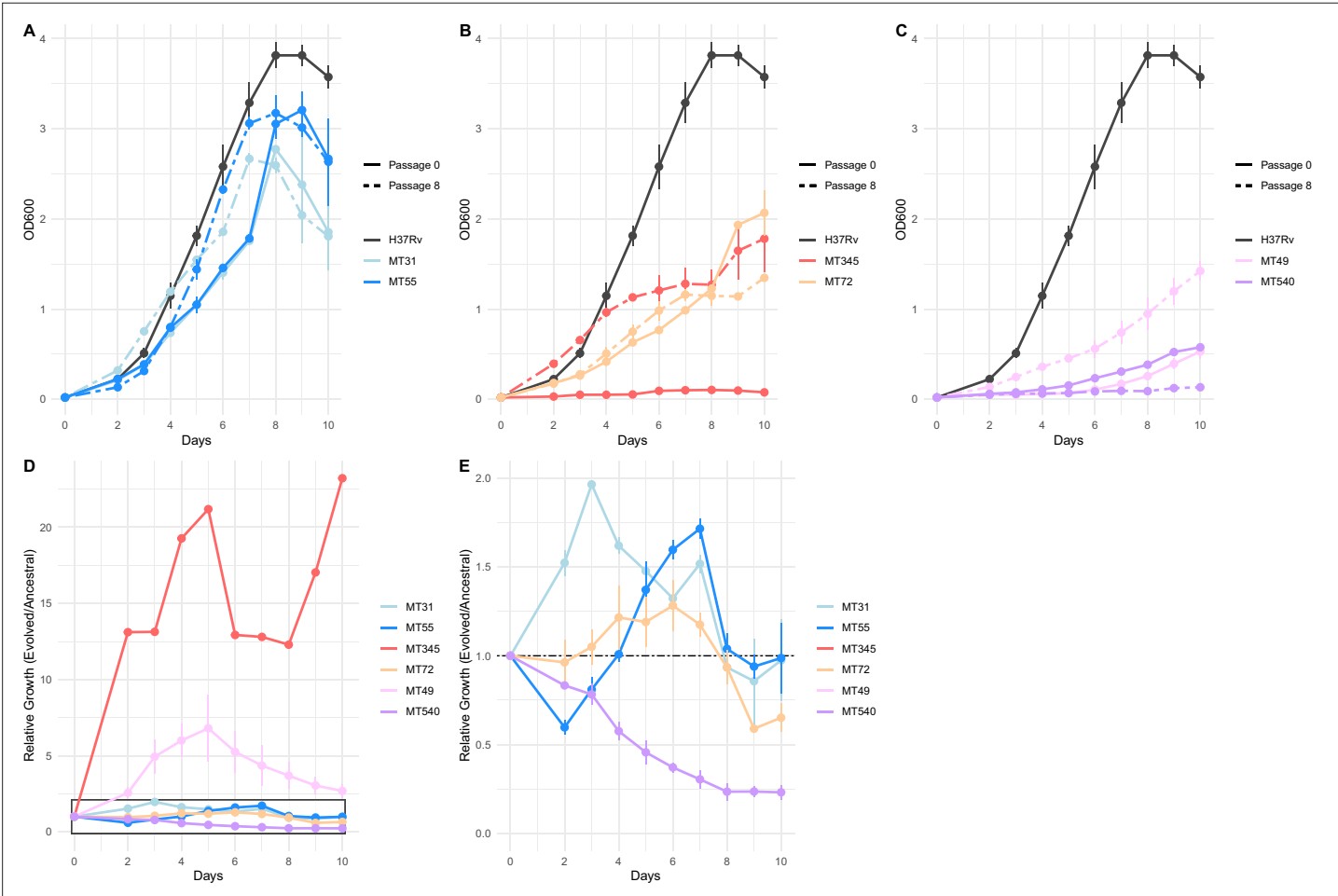

**Figure 4.** Impacts of pellicle passaging on planktonic growth rate vary by genetic background. (**A, B, C**) Planktonic growth and (**D, E**) relative fitness curves comparing the growth of strains after eight pellicle passages to the ancestral strains. Growth curves were performed in triplicate, and the mean $OD_{600}$ value is plotted with error bars representing ±1 SD. Relative fitness was calculated at each timepoint as $OD_{600}$ Passage 8 / $OD_{600}$ Passage 0. Panel E shows boxed region from panel D in more detail.

The online version of this article includes the following source data for figure 4:

**Source data 1.** Planktonic growth curves of ancestral and pellicle evolved strains.

### Fixed Mutations by Strain

| Strain | Position* | Frequency at passage 0 | Fixed at Passage | Type | Mutation | Gene | Product | Frequency in Global Dataset |
|---|---|---|---|---|---|---|---|---|
| **L4.9** | | | | | | | | |
| MT31 | 2279599 | 40% | 8† | Synonymous | D157D (GAT-GAC) | acg | conserved hypothetical, possible nitroreductase | 0 |
| MT31 | 2447977 | 39% | 8 | Synonymous | I6I (ATC-ATT) | Rv2186c | conserved hypothetical | 0.0025% |
| **MT31** | **3553976** | — | **8** | **~120 kb DUP** | — | **Rv3188 [. . .] Rv3296** | — | **NA** |
| MT31 | 4115505 | 49% | 20† | Synonymous | P116P (CCG-CCT) | nth | endonuclease III | 0.010% |
| **MT55** | **3589531** | — | **4** | **~120 kb DUP** | — | **Rv3212 [. . .] Rv3324c** | — | **NA** |
| **L4.4.1.2** | | | | | | | | |
| MT345 | 580968 | 1.50% | 4 | Non-synonymous | M54V (ATG-GTG) | regX3 | two component sensory transduction protein RegX | 0 |
| MT345 | 852317 | <1% | 4 | Non-synonymous | R237L (CGC-CTC) | phoP | two component system response transcriptional positive regulator PhoP | 0 |
| MT345 | 3876836 | 0% | 8 | Intergenic | -14/+54 (C-A) | truA / rplQ | tRNA pseudouridine synthase A / 50S ribosomal protein L17 | 0 |
| MT72 | 499768 | 5.70% | 4 | 1 bp DEL - frameshift | pos. 56/654 | renU (mutT3) | nudix hydrolase | 0 |
| MT72 | 652668 | 9.10% | 4 | Non-synonymous | R30S (CGC-AGC) | Rv0561c | oxidoreductase | 0.0025% |
| MT72 | 1416232 | 30.00% | 16† | Synonymous | C372C (TGT-TGC) | embR | transcriptional regulator | 0 |
| MT72 | 4168394 | 0% | 4 | 1 bp DEL- noncoding | pos. 50/86 | serV | tRNA-Ser (GGA) | 0 |
| **L4.4.1.1** | | | | | | | | |
| MT49 | 1939638 | 55% | 8† | Non-synonymous | P14T (CCG-ACG) | cmk | cytidylate kinase | 0 |
| MT49 | 2799039 | 0% | 8 | Synonymous | G614G (GGC-GGT) | Rv2488c | LuxR family transcriptional regulator | 0.0025% |
| **MT49** | **3690947** | **0%** | **8** | **Intergenic** | **-9/-194 (A-G)** | **lpdA / Rv3304** | **NAD(P)H quinone reductase / hypothetical** | **0.15%** |
| MT540 | 648780 | 6.00% | 4 | Non-synonymous | F82S (TTC-TCC) | mgtA | GDP-mannose-dependent alpha-mannosyltransferase | 0 |
| MT540 | 3423753 | 0% | 4 | Non-synonymous | D559Y (GAT-TAT) | fadE22 | acyl-CoA dehydrogenase FadE22 | 0 |
| **MT540** | **3690952** | **0%** | **8‡** | **Intergenic** | **-14/-189 (G-C)** | **lpdA / Rv3304** | **NAD(P)H quinone reductase / hypothetical** | **0.010%** |

* Coordinates in H37Rv genome
† Mutation fell out of population, fixed = 0%
‡ At 85% frequency at passage 8

**Figure 5.** Small number of variants that became fixed (>95 or <5% frequency) throughout the course of the experiment. Including both single nucleotide polymorphisms and insertions/deletions (INDELs) identified by Popoolation2, Breseq, and/or Pilon. Highlighted in bold are two instances of convergent adaptation we observed within the L4.9 and L4.4.1.1 sub-lineages. These mutations are exclusive to strains passaged as pellicles, see *Supplementary file 2* for mutations arising in strains passaged planktonically.

## Convergent evolution of large duplication

We discovered a large (~120 kb) duplication in pellicles of both strains from lineage L4.9 (MT31 and MT55, *Figure 5*) following pellicle passaging. The duplications were initially identified with Pilon (*Walker et al., 2014*), and to confirm their coordinates and absence in the ancestral populations, we made sliding window coverage plots in this region and confirmed ~2× coverage indicating the presence of a duplication (*Figure 8*). The coordinates of the duplication appear to be slightly different in the two populations with the duplication in MT55 starting about 3 kb after the duplication in MT31 (*Figure 8*). We determined that the duplication was fixed after the first sequenced passage in each population (passage 4 and 8 for MT55 and MT31, respectively; *Supplementary file 1*). After fixation, the duplication appears to have remained stable in both populations, as far out as 20 passages in MT31, as we do not see any reduction in coverage across or within the duplicated region at these later passages (*Figure 8*). This duplication did not emerge in planktonically passaged isolates of MT31 and MT55, suggesting it has some specificity for pellicle adaptation of these strains (*Figure 7—figure supplement 1*).

There are a total of 172 genes within the duplicated regions in MT31 and MT55 (*metU* through *folD*). To identify a possible functional enrichment within these genes, we performed a GO term

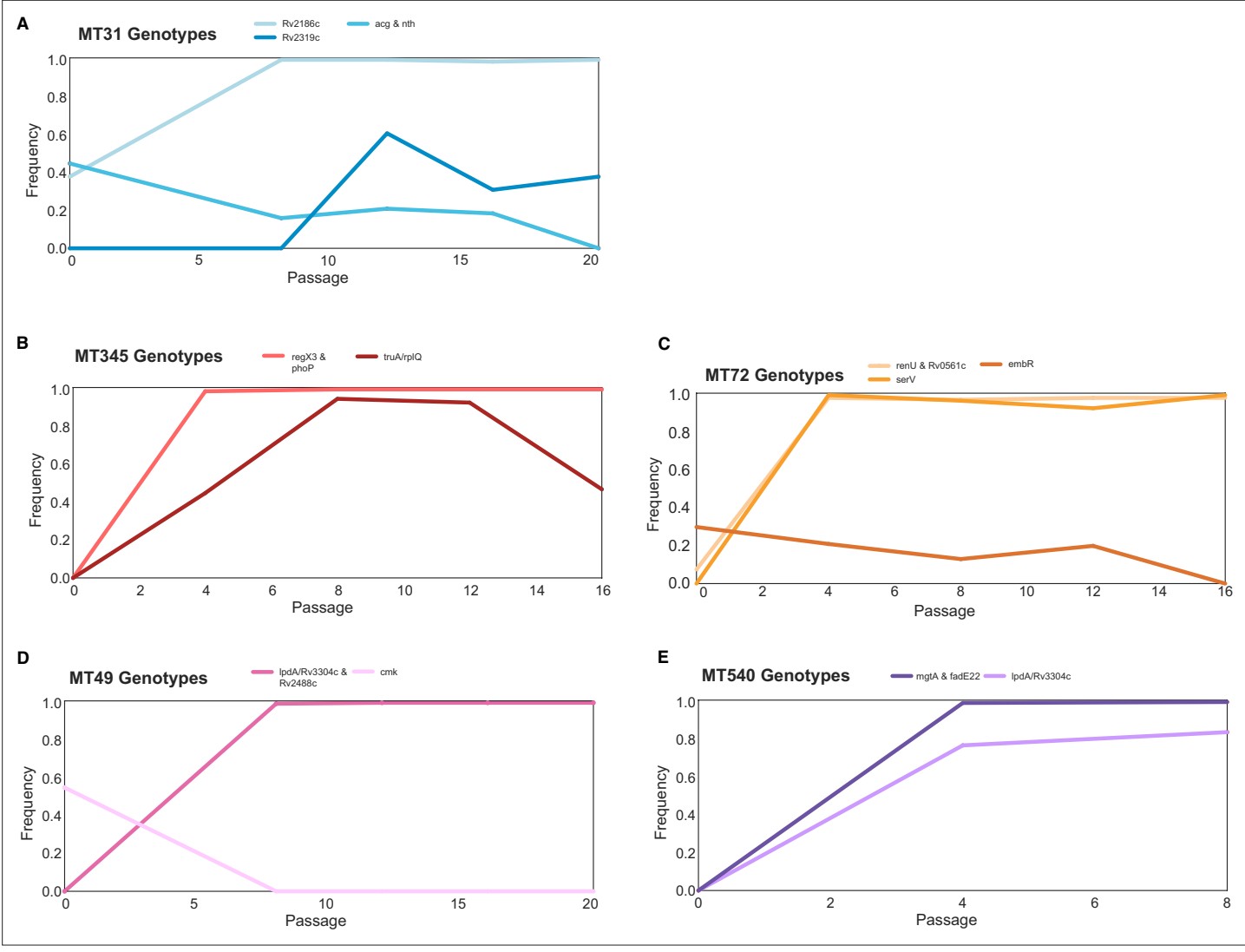

**Figure 6.** Trajectories of mutations that changed ≥30% over the course of the experiment show few mutations arose during passaging. The mutations that became fixed almost always did so in the first or second sequenced passage (***Supplementary file 1***), and we identified very few mutations with substantial declines in frequency. Mutation frequency data were calculated with Popoolation2 and plots were made using Lolipop (see Methods).

The online version of this article includes the following figure supplement(s) for figure 6:

**Figure supplement 1.** Population genetics statistics calculated from Pool-seq data from all strains at all sequenced passage points.

**Figure supplement 2.** Trajectories of mutations that changed ≥30% over the course of planktonic passaging.

enrichment analysis and compared our results to the results from randomly selected groups of contiguous genes from throughout the genome (see Methods). While the initial analysis identified significant enrichment of genes involved in nucleotide metabolism within the duplication (***Supplementary file 3***), this enrichment does not appear significant when compared with randomly selected gene sets (***Figure 8—figure supplement 2***).

## Regulatory pathways involved in biofilm formation

Having identified two instances of convergent evolution at the level of individual loci, we investigated further for evidence of convergent evolution at the regulatory pathway level. We identified 14 genes of interest (GOI) in association with variants that rose to fixation during the passaging experiment (***Figure 5***). Using the MTB Network Portal v2 database of transcription factor overexpression (TFOE) and ChIP-Seq data (http://networks.systemsbiology.net/mtb/), we identified regulatory pathways that contained GOI. In ***Figure 9***, we show the TFs that are predicted to affect the expression of multiple

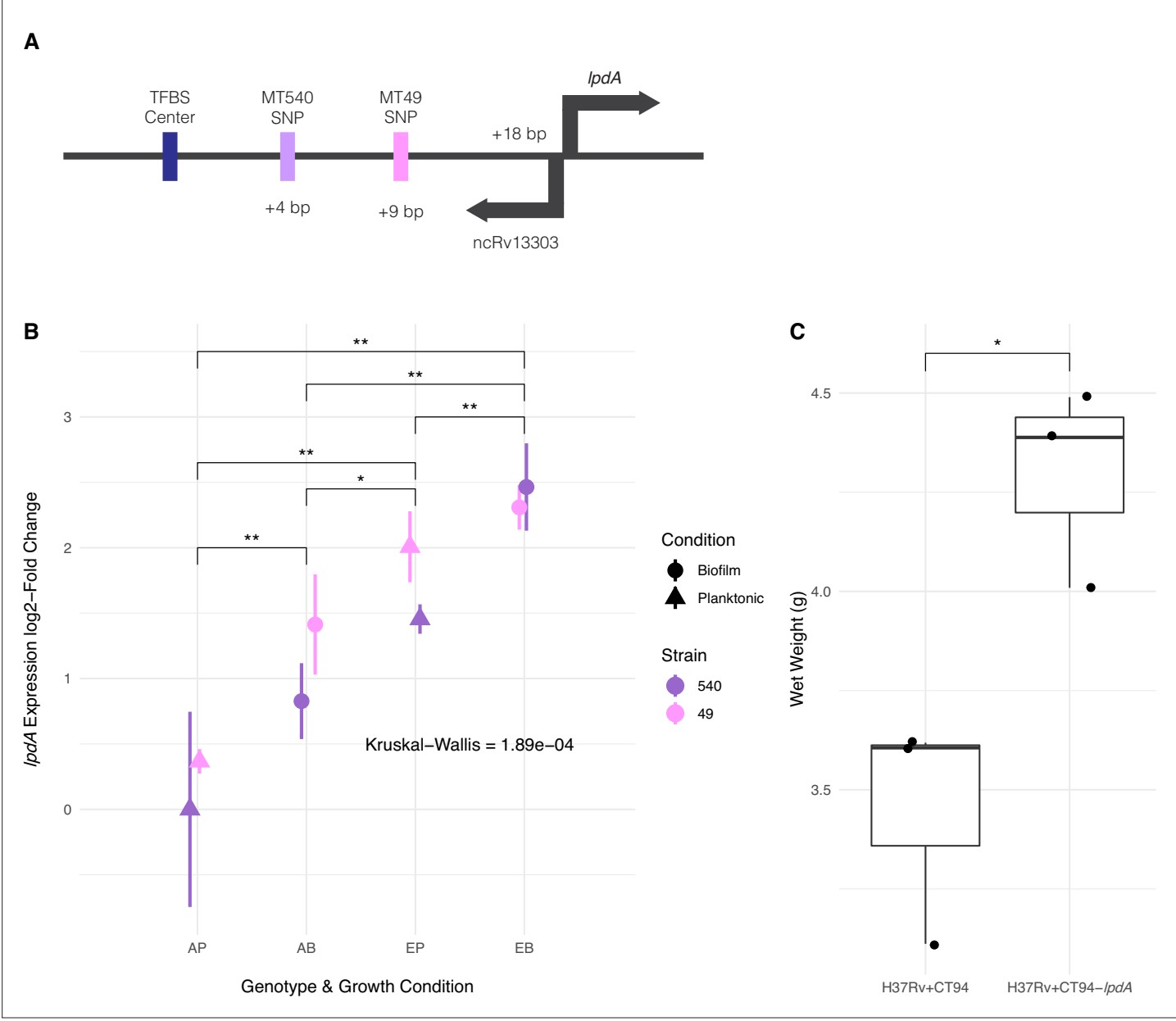

**Figure 7.** Adaptation to pellicle growth via increased expression of lpdA. (**A**) Diagram of convergent adaptation in L4.4.1.1 strains MT540 and MT49 in the region upstream of *lpdA* within a transcription factor binding site (TFBS) and non-coding RNA. (**B**) Expression of *lpdA* differed significantly between sample groups (Kruskal–Wallis test, p-value=1.89e-04). Pairwise comparisons between groups (n=3, Mann-Whitney U test with Benjamini-Hochberg correction) revealed significantly increased expression in evolved (E) strains when compared with ancestral (A) strains, as well as during biofilm (B) growth when compared with planktonic (P) growth. (**C**) Overexpression of *lpdA* in H37Rv results in significantly (n=3 for each construct, Mann-Whitney U test) increased biofilm wet weights. p-value legend: *p<0.05, **p<0.01.

The online version of this article includes the following source data and figure supplement(s) for figure 7:

**Source data 1.** RT-qPCR of *lpdA* (raw and log-2 fold changes) and pellicle wet weights from *lpdA* overexpression constructs shown in *Figure 7B–C*.

**Figure supplement 1.** H37Rv pellicles with empty vector (left) and extra copy of *lpdA* (right) introduced into the chromosome using pCT94.

GOI. Of our GOI, *acg* is the most highly regulated, with seven TFs predicted to affect its expression. The function of *acg* is unknown: it is annotated as a conserved hypothetical, but according to in vivo studies may play an important role in infection (*Hu and Coates, 2011*; *Singh et al., 2019*). Additionally, we used these data to determine if GOI annotated as regulators (*regX3*, *phoP*, *Rv2488c*, and *embR*) regulated other GOI (*Figure 9*). We found again that *acg* appears to be highly regulated, and

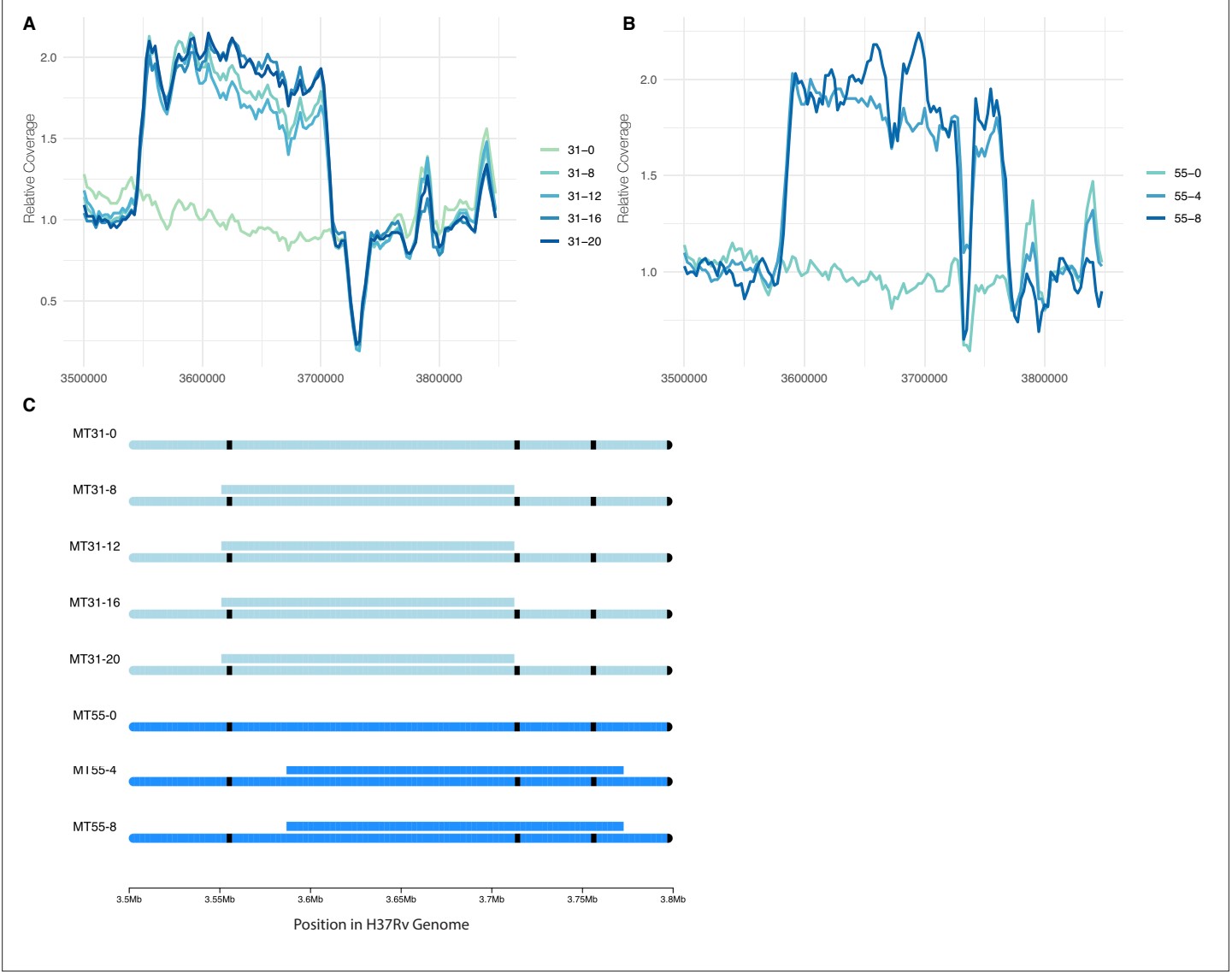

**Figure 8.** Convergent evolution of a large duplication in L4.9 strains MT31 and MT55, within the same genomic region flanked by mycobacterial IS6110 insertion sequences. Sliding window coverage plots show the increase in coverage in this region for strain (**A**) MT31 and (**B**) MT55, and a chromosome plot (**C**) shows the duplications' coordinates and overall stability in both populations over multiple passages along with the location of IS6110 insertion sequences in black.

The online version of this article includes the following figure supplement(s) for figure 8:

**Figure supplement 1.** Sliding-window plots showing relative sequencing coverage of planktonically passaged isolates (n=3 independently evolving replicates per strain).

**Figure supplement 2.** Results of 100 permutations of random gene ontology (GO) term enrichment analysis (see Methods).

that *phoP* also regulates *mgtA* and *fadE22*, which are both hypothesized to have a role in cell-wall processes. Taken together these results indicate that *M. tb* has a highly complex and tightly regulated network and suggests that biofilm formation involves a complex network of interacting loci.

## Discussion
### Variation within the *Mycobacterium tuberculosis* complex (MTBC)
It has long been recognized that the MTBC is divided into well differentiated, globally circulating lineages (*Hirsh et al., 2004*). As an obligate human pathogen, *M. tb* is subject to the vagaries of human

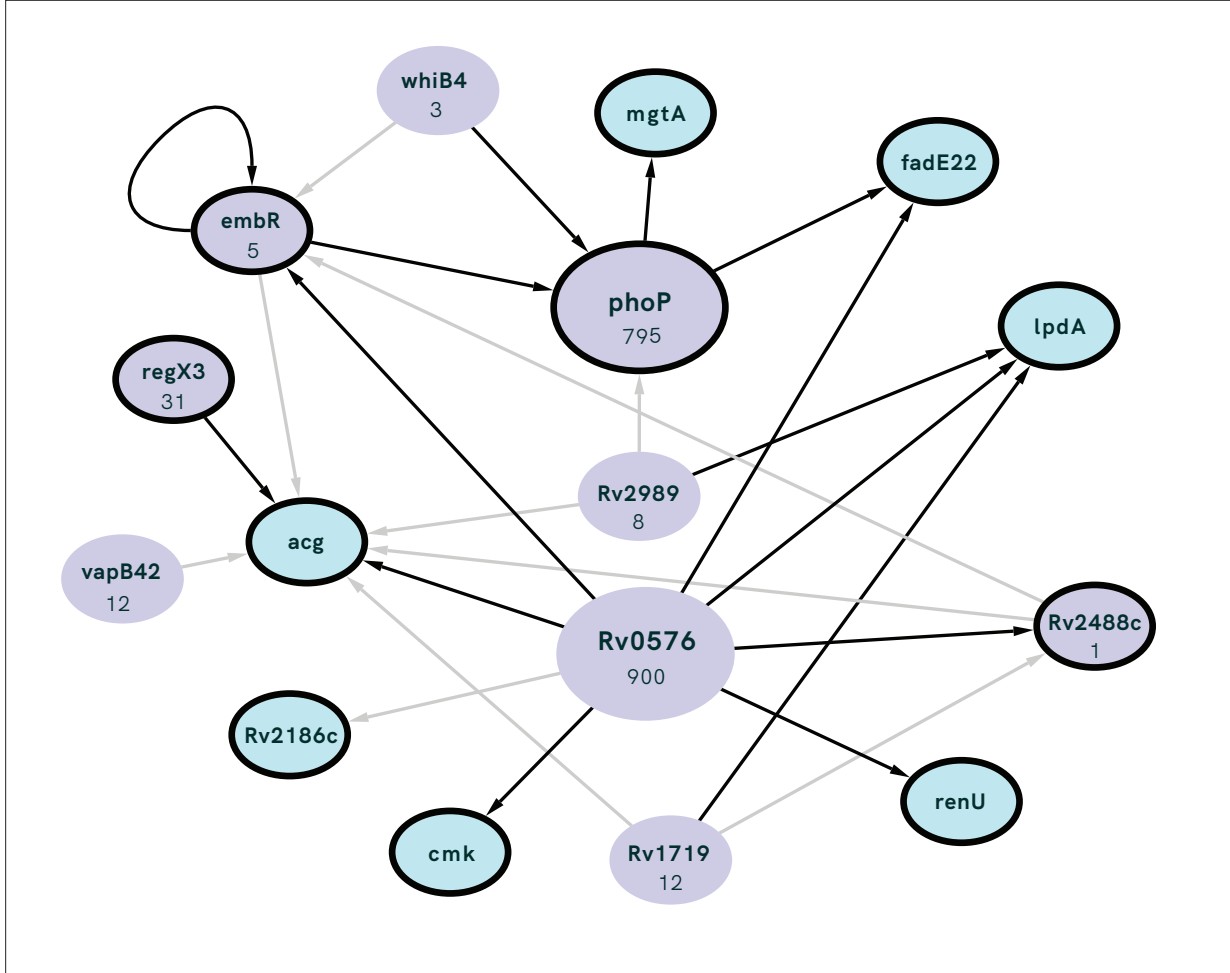

**Figure 9.** Interconnected regulation of our genes of interest (GOI; outlined in black) by common transcription factors (TFs; purple) with the number of genes regulated by each TF listed under gene name. TF identified as having significant overlap with our GOI (i.e. the number of GOI regulated by the TF is higher than expected by chance) using the MTB Network Portal (see Methods). Edges between nodes indicate >0.5 log2-fold absolute change in expression in a transcription factor overexpression strain, black lines show statistically significant (p<0.05) differential expression.

behavior, and this behavior has had a prominent role in structuring *M. tb* populations (*Pepperell et al., 2011*; *Brynildsrud et al., 2018*; *Liu et al., 2018*; *Mulholland et al., 2019*; *O'Neill et al., 2019*). In addition to genetic variation, phenotypic differences have been described among strains of *M. tb*. Differences have been observed in a range of phenotypes including metabolism (*Øyås et al., 2020*), gene expression (*Domenech et al., 2017*), DNA methylation (*Modlin et al., 2020*), in vitro stress responses (*Tizzano et al., 2021*), and biofilm morphotype (*Pang et al., 2012*).

We have shown previously that *M. tb* populations are genetically sub-divided over fine scales as a result of neutral genetic drift (*Pepperell et al., 2010*). Here, we find that *M. tb* strains from within a single lineage (lineage 4, 'Euro-American'), separated by few genetic differences (*Figure 1*), exhibit variation across a range of phenotypes. Prior to the imposition of selection pressures in vitro, differences among strains were evident in pellicle morphotype (*Figure 1*), cell length (*Figure 2*), pellicle wet weight (*Figure 3*), and planktonic growth rate (*Figure 4*). In some cases, phenotypic clustering occurred at the sub-lineage scale – e.g., planktonic growth curves of L4.9 isolates – and in others sub-lineage strain pairs exhibited marked differences – e.g., planktonic growth rate and pellicle wet weight of the L.4.1.1.2 strains MT345 and MT72.

## Genetic and phenotypic responses to selection for biofilm growth

Regardless of phenotypic starting point, all strains of *M. tb* in our study showed evidence of adaptation in response to selection for pellicle growth. The gross morphology of *M. tb* pellicles changed over

the course of the experiment (*Figure 1*), pellicle wet weights increased for all strains (*Figure 3*) and increases in the amount of ECM were evident in SEM for all strains (*Figure 2*). MT345 formed robust pellicles at baseline yet manifested a dramatic increase in wet weight following pellicle passaging, indicating that adaptive capacity remains even among high biofilm forming clinical isolates (*Figure 3*). Collectively, these results demonstrate that our method was effective in exerting selection for pellicle growth. More broadly, it shows that experimental evolutionary approaches can be applied successfully to complex phenotypes and fastidious, slow growing, and non-model organisms.

During our experiment, pellicle phenotypes stabilized between passage 2 and 8 (*Figure 1*); extended passaging beyond eight passages did not produce further phenotypic changes (*Figure 1— figure supplement 1*). Fixation of novel and previously segregating mutations occurred over the same time frame (*Figure 5*; *Figure 6*), and we did not observe any further sweeps over the course of the experiment. Within eight passages, we thus saw phenotypic and genotypic stabilization across all populations. This phenomenon is frequently observed during experimental evolution, where rapid adaptation is followed by diminishing returns after continued passaging (*Barrick et al., 2009*; *Khan et al., 2011*; *Philippe et al., 2007*).

We observed few mutations over the course of the experiment (*Figure 5*), and some of these are likely to be hitchhikers linked to advantageous alleles that swept to fixation (*Figures 5 and 6*). These few mutations appeared to have pleiotropic effects, as we observed changes in phenotypes other than the one under selection. 'Off target' phenotypes, such as planktonic growth rate and cell length, varied among strains. Experimentally evolved populations generally manifest first-pass, large-effect mutations during initial adaptation; our results suggest that these mutations in *M. tb* are broad in their scope and variable in their effects. The latter phenomenon may be due to a large genetic target for biofilm growth, the impacts of hitchhiking mutations, and/or genetic background of the strains.

Candidate SNPs and deletions mediating pellicle adaptation were either exceedingly rare or absent in our search of a database of almost 40,000 *M. tb* genomes. This suggests that the loci hit in our experiment are under strong purifying selection in *M. tb*'s natural environment: that is, the mutations we observed are associated with substantial fitness costs to the bacterium during human infection. This is not to suggest that biofilm growth is irrelevant to natural infection; as discussed above, *M. tb* biofilms and biofilm biomarkers have been identified in clinical specimens (*Canetti, 1955*; *Nyka, 1977*; *Nyka, 1967*; *Nyka, 1963*; *Nyka and O'Neill, 1970*; *Chakraborty et al., 2021*). Rather, we hypothesize that the mutations we observed are selected against because they interfere with regulatory pathways that normally constrain biofilm growth to specific environmental conditions. Regulatory constraints would limit potential drawbacks of biofilm formation, such as the metabolic costs of matrix production and immunostimulation. Another, not mutually exclusive, explanation is that fitness costs associated with mutations observed here are due to effects on off-target traits such as planktonic growth rate, cell size changes, and/or phenotypes unmeasured here. Experiments such as this one can illuminate impermissible paths in *M. tb* adaptation and reveal specific vulnerabilities with the potential to be exploited in TB therapeutics.

Two strains in our study evolved a large (~120 kb) duplication in the same genomic region, which fixed by passage 8 (*Figures 5 and 8*). The duplication persisted without evidence of decay in extended passaging in our study (up to passage 20 in MT31). Duplications in this region of the genome have been identified previously in isolates from lineages 2 and 4 (*Domenech et al., 2010*; *Domenech et al., 2014*; *Weiner et al., 2012*). Duplications at this site are likely facilitated by flanking *IS6110* elements (*Figure 8C*) which can induce unequal homologous recombination. Prior research has shown that the duplication confers a fitness disadvantage in vivo (as measured in a mouse model) and can be selected during axenic culture (*Domenech et al., 2014*).

Here, we find this lab adaptation to be associated with enhanced pellicle formation. Interestingly, while the duplication was associated with increases in planktonic growth rate, we did not identify it in planktonically passaged populations (*Figure 4*, *Figure 7—figure supplement 1*). *Domenech et al., 2014* found previously that emergence of a similar duplication was variable among strains and culture conditions. Together, these data suggest that while the duplication is observed commonly in the lab, it is selected on specific genetic backgrounds and in response to specific environmental conditions. Large-scale genome rearrangements including duplications have previously been identified as a rough adaptive 'first draft' followed by further refinement during experimental evolution (*Lynch and Conery, 2000*; *Zhang, 2003*). The duplication was the only mutation that fixed in MT55, which exhibited a

dramatic increase in pellicle weight (*Figure 3*) and increased production of ECM (*Figure 2*) following passaging, suggesting that it is indeed this seemingly multipurpose mutation that is responsible for pellicle adaptation. The apparent impacts of the duplication were not limited to pellicle growth, as MT55 also exhibited an increase in cell length (*Figure 2*) and a faster rate of growth during planktonic culture, as did its sister strain MT31 also encoding the mutation (*Figure 4*). Taken together, our findings suggest that the production of ECM and other aspects of biofilm growth are tightly regulated during natural infection, and that first pass mutations outside of this environment result in de-repression of biofilm production. Our findings further suggest that biofilm formation is co-regulated with a range of phenotypes, consistent with prior research on biofilms, persistence and drug tolerance (*Richards et al., 2019*).

## Epistasis

The 120 kb duplication occurred on two strains from the same sub-lineage (MT31 and MT49 from L4.9) and did not occur in any other genetic background. A similar phenomenon emerged in association with sub-lineage L4.4.1.1, where both strains underwent mutation at a TFBS upstream of *lpdA*, with attendant impacts on gene expression. While our strains are all very closely related, there are still distinct genetic variants for each sub-lineage that may affect the evolutionary trajectory and lead to convergent adaptation of strains within them (*Supplementary file 4*). We observed similar patterns in the planktonically passaged populations, for example in repeated mutations occurring in an apparently sub-lineage (*tuf*, L4.1.1.2) or strain-specific manner (*Supplementary file 2*, *Figure 6—figure supplement 2*). Repeated mutation at these loci provides strong support for their involvement in the trait under selection. In addition, the association of candidate adaptive mutations with specific sub-lineages suggests that *M. tb* strains' genetic backgrounds played a role in shaping the adaptive path to enhanced biofilm growth.

Epistasis, which refers to the phenotypic impact of interactions among mutations at distinct loci, has been observed previously during experimental evolution of microbes (e.g. *Fisher et al., 2019*). Sign epistasis is a form of epistasis in which a mutation is advantageous on one genetic background and deleterious on another (*Weinreich et al., 2005*). The presence of sign epistasis can hinder access to a fitness optimum, and thus it represents a constraint on adaptation (*Weinreich et al., 2005*; *Poelwijk et al., 2007*). Our identification of mutations only in association with specific sub-lineages is consistent with sign epistasis, as is the observation of parallel adaptation in this small sample of strains: it suggests, for example, that mutations upstream of *lpdA* may be deleterious on some genetic backgrounds, and/or that alternatives are deleterious on the L.4.1.1.1 background. A careful review of the existing literature reveals additional evidence of sign epistasis in *M. tb*. For example, we and others have found evidence suggesting that *M. tb* lineage affects the mutational path to drug resistance (*Mortimer et al., 2018*; *Farhat et al., 2019*; *Castro et al., 2020*). Even more striking, Manson et al. identified a stereotypic order of resistance mutation acquisition in a large survey of drug-resistant *M. tb* isolates, a finding that was subsequently replicated (*Manson et al., 2017*; *Ektefaie et al., 2021*). This suggests that the acquisition of an initial resistance mutation alters the fitness associated with subsequent mutations, as has been observed in other settings (*Silva et al., 2011*). More direct evidence for sign epistasis comes from mutagenesis experiments, in which the lethality of gene knockouts has been shown to vary according to *M. tb* strain genetic background (*Maksymiuk et al., 2015*; *Carey et al., 2018*).

## Upregulation of *lpdA* during biofilm growth

As noted above, we identified convergent adaptation in the region upstream of *lpdA* (*Rv3303c*) on L.4.1.1.1 backgrounds. *lpdA* is annotated as a NAD(P)H quinone reductase, which is known to be expressed in vivo (*Deb et al., 2002*) and to contribute to virulence in mice (*Akhtar et al., 2006*). It has not previously been linked with biofilm formation. The mechanism of its contribution to virulence in vivo is thought to be by protection against oxidative stress (*Akhtar et al., 2006*). The strains in our study evolved variants 5 bp apart, both of which lie within the TFBS for *Rv1719* (*Figure 7A*). These SNPs also lie within ncRNA *ncRv13303c*, which was originally identified by RNA sequencing (*Arnvig et al., 2011*; *DeJesus et al., 2017*) but is yet uncharacterized. Previous work characterizing variable-number tandem repeats (VNTR) in this intergenic region have proposed a possible hairpin structure that lies within *ncRv13303c* and could affect its function (*Zheng et al., 2008*). Differences in VNTR

copy number and promoter sequence in this region have been observed between different strains of mycobacteria, and even linked to differences in expression of *lpdA* (*Akhtar et al., 2009*; *Pérez-Lago et al., 2013*; *Zheng et al., 2008*).

We hypothesized that SNPs in the TFBS upstream of *lpdA* would affect TF binding and expression of *lpdA*. This was indeed the case, as evolved strains had significantly higher expression of *lpdA* (*Figure 7B*). Additionally, we found expression of this gene to be higher during biofilm growth when compared with planktonic growth. Further support for a link between increased expression of *lpdA* and enhanced pellicle growth comes from our data showing an increase in pellicle wet weight following introduction of a second copy of the gene via an integrative plasmid (*Figure 7C*). Interestingly, *lpdA* is within the bounds of the large duplications we identified in strains MT31 and MT55, which may affect the dosage of this gene in those evolved strains.

The two SNPs upstream of *lpdA* were exceedingly rare among clinical isolates of *M. tb*: in a search of almost 40,000 *M. tb* genomes, we identified the MT49 SNP in just 61 genomes and the MT540 SNP was found in 4 genomes (*Figure 5*). Interestingly, the MT49 mutation was shown to sweep to fixation in a deep sequencing study of serial sputum samples chronicling *M. tb* within host dynamics (*Trauner et al., 2017*). This supports the hypothesis that biofilms are important at a specific phase of TB infection, and that enhanced biofilm growth can also occur in vivo through increased expression of *lpdA* and genes co-regulated with it. Taken together, these observations suggest that variants at the TFBS upstream of *lpdA* can be transiently selected during infection but are otherwise deleterious, possibly due to fitness costs associated with biofilm growth that is timed inappropriately.

## The regulatory system as a target of selection in *M. tb*

The L4.9 duplication and SNPs upstream of *lpdA* presumably act via regulation/gene dosage. Regulatory activity was also common among the other loci that accumulated mutations in our experiment, including phosphorelay signal transduction and DNA-binding activity (*Figure 5*). We identified fixed SNPs in four transcriptional regulators: *regX3*, *phoP*, *embR*, and *Rv2488c* (*Figure 5*). These genes, in turn, regulate other candidate genes identified in our study (*Figure 9*). *regX3* and *phoP* belong to two-component regulatory systems (2CRS), whereas *embR* and *Rv2488c* are of the ompB and luxR families, respectively. Many of *M. tb*'s 2CRS have been implicated in virulence and survival in host conditions (*Li et al., 2019*) and this is true of *regX3*, which with its partner *senX3* is required for virulence in mice (*Li et al., 2019*; *Parish et al., 2003*). Of note, *regX3* regulates genes that are known to be involved in *M. tb* biofilm development (*Richards et al., 2019*). Similarly, while *Rv2488c* has not previously been directly implicated in *M. tb* biofilm growth, *luxR* regulators are involved in quorum sensing in gram-negative bacteria and have been hypothesized to play a similar role in mycobacteria (*Chen and Xie, 2011*; *Sharma et al., 2014*). Our results provide support for further investigation of quorum sensing in *M. tb* and its relationship to biofilm development.

Several studies have demonstrated that clinical isolates of *M. tb* vary in their methylation patterns, and methylome phenotypes have been linked with allelic variants in cognate methyltransferases (*Modlin et al., 2020*; *Phelan et al., 2018*; *Shell et al., 2013*; *Zhu et al., 2016*). Methyltransferase motifs are associated with regulatory elements such as promoters, and variation in methylation patterns has been associated with differences in gene expression in some cases (*Chiner-Oms et al., 2019*; *Gomez-Gonzalez et al., 2019*; *Modlin et al., 2020*). We did not observe any mutations at loci known to encode methyltransferase genes, but it is possible that adaptation to pellicle growth was accompanied by epigenomic changes, or that epigenomic changes occurring by an unknown mechanism contributed to the phenotypic changes we observed. Such epigenomic changes would have to be relatively stable, as the phenotypic changes we observed persisted following growth under standard planktonic conditions.

Broadly speaking, our results show that selection in our system appears to have targeted genes that regulate many other genes (hubs, e.g. *phoP*), and genes under regulation by many genes (e.g. *acg*), whose expression is likely responsive to a broad range of conditions. This reflects general features of *M. tb*'s regulatory network, which exhibits a high degree of connectivity between regulatory pathways (*Chauhan et al., 2016*; *Galagan et al., 2013*) as well as a hierarchical structure in which master regulators can rapidly calibrate global patterns of gene expression (*Chauhan et al., 2016*; *Parvati Sai Arun et al., 2018*). Similar findings have also been obtained in other systems, where experimentally evolved populations have shown adaptation in TFs and other global regulators,

with global impacts on gene expression (*Ali and Seshasayee, 2020*; *Conrad et al., 2010*; *Philippe et al., 2007*; *Rodríguez-Verdugo et al., 2016*; *Saxer et al., 2014*). These mutations occur quickly to enable rapid adaptation to new environments, and it is hypothesized that secondary selection may act to refine gene expression after the initial burst of adaptation (*Ali and Seshasayee, 2020*; *Rodríguez-Verdugo et al., 2016*). *Thorpe et al., 2017* previously found evidence of positive selection at promoter sites in natural populations of *M. tb*, suggesting that adaptation in natural as well as experimental populations is facilitated by regulatory mutations. The broad potential impacts of regulators such as *regX3*, *phoP*, *embR*, and *Rv2488c* and large-scale duplications likely helps to explain the wide range of phenotypic changes we observed. Further passaging of our isolates will reveal whether these adaptations can be further refined or whether pellicle growth remains inextricably linked to a range of phenotypes.

Of note, none of the genes identified in our study have previously been implicated in biofilm development, except for *renU/mutT3*. Interestingly, previous research (*Wolff et al., 2015*) found that *renU* knockouts were defective in pellicle growth, whereas, in our study, a frameshift mutation in *renU* was associated with enhanced biofilm growth (MT72). The novelty of our findings highlights the potential for experimental evolution to complement more traditional approaches of interrogating important pathogen phenotypes, particularly in elucidating complex interactions and the phenotypic impacts of subtle genetic changes.

From some perspectives, the adaptation of *M. tb* seems impossibly constrained: subject to genetic drift imposed by its host's unpredictable behavior, subject to the limitations of a fully linked genome, and navigating a fitness landscape rendered complex by powerful epistatic interactions. Yet *M. tb* adapts rapidly in both natural and experimental settings. *M. tb*'s complex regulatory architecture exhibits many of the features that characterize robust systems – such as redundancy, modularity, and multiple feedback mechanisms (*Kitano, 2004*) – and we hypothesize that it is among the reasons *M. tb* continues to persist and thrive across the globe.

## Materials and methods
### Bacterial strains and growth conditions

Clinical strains were initially isolated from sputum samples. Briefly, sputum samples were de-contaminated and struck on Löwenstein–Jensen slants. *M. tb* growth was inoculated into Middlebrook 7H9 broth (HiMedia) containing 0.2% w/v glycerol, 10% v/v OADC supplement (oleic acid, albumin, D-glucose, and catalase; Becton Dickinson), and 0.05% w/v Tween-80 (7H9OTG) and incubated at 37°C with 5% $CO_2$ for 3–5 weeks. Cultures were sub-cultured in 7H9OTG prior to the start of this experiment. Six clinical strains were selected for experimental passage (MT31, MT49, MT55, MT72, MT345, and MT540). The terminal culture (passage 0) of each strain was grown to an $OD_{600}$ ~1 in planktonic culture and inoculated into media for biofilm growth (passage 1; *Figure 10*). Aliquots of passage 0 were frozen, and gDNA was isolated for sequencing. For planktonic growth, 250 μL was inoculated into 5 mL 7H9OTG, in duplicate, and incubated at 37°C with shaking, until grown to an $OD_{600}$ ~1. For biofilm growth, 250 μL of planktonic culture was inoculated into 25 mL Sauton's medium (for 1 L: 0.5 g $KH_2PO_4$, 0.5 g $MgSO_4$, 4 g L-asparagine, 2 g citric acid, 0.05 g ammonium iron (III) citrate, 60 mL glycerol, adjust pH to 7.0 with NaOH) containing 0.1% w/v $ZnSO_4$, in duplicate, and incubated at 37°C, with 5% $CO_2$, without shaking for 5–7 weeks. Biofilms were grown as previously described (*Kulka et al., 2012*): 250 mL bottles (Corning, 430281) were incubated with a tight-fitting cap for 3 weeks, and then with a loose-fitting cap for the remainder of growth.

### Experimental evolution of pellicles

As illustrated in *Figure 10*, duplicate populations of six clinical strains were grown as a biofilm. After 5–7 weeks, the more robust of the two pellicles was selected for passage. About 0.3 g (wet weight) of pellicle was inoculated into 25 mL Sauton's in duplicate and grown as previously described. The pellicle not selected for passage was discarded. Between 8 and 20 total pellicle passages per strain were performed. Every four passages culture were frozen at –80°C and gDNA was extracted for sequencing (excluding MT31 and MT49, for whom the first passage that was frozen and had gDNA extracted was passage 8; *Supplementary file 1*).

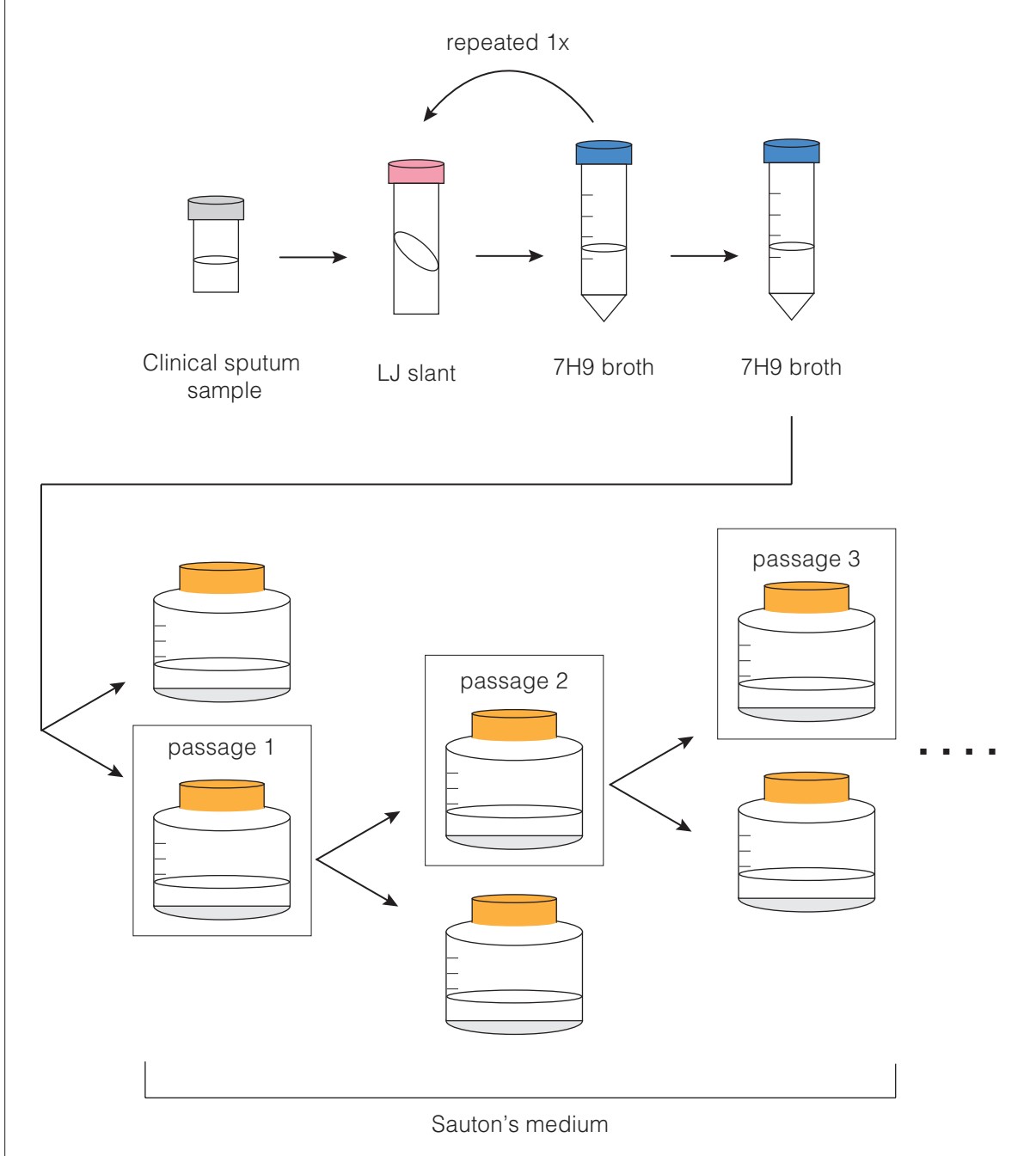

**Figure 10.** Protocol for serial passage of *M. tb* pellicles.

## Experimental evolution of planktonic cultures

Starter cultures of ancestral strains were made from frozen cryovial stocks and sequenced to get an ancestral baseline for the planktonic passages. Each strain was inoculated in triplicate to form three independently evolving populations: A, B, and C. For each passage, 25 mL of 7H9OTG was inoculated with 500 μL of culture at $OD_{600}=1$. Cultures were grown at 37°C with shaking until an $OD_{600}$ ~1, before freezing cryovials, extracting gDNA for sequencing, and seeding the next passage. A total of 15 of 18 of the populations were passaged four times, while 3 of 18 were only passaged three times due to the presence of contamination.

## Pellicle phenotypes and wet weights

Pellicle growth was photo documented and phenotyped after 5–7 weeks of growth during the course of the experiment. Wet weights were taken after regrowing both ancestral and passage 8 strains from cryovial stocks, as described above, inoculating first into 7H9OTG and then into Sauton's either in duplicate or triplicate. Wet weight measurements were taken after 5 weeks of pellicle growth by removing the spent media beneath the pellicle and weighing. Then, the tare weight of the empty bottle was subtracted.

## Scanning electron microscopy

For SEM experiments, samples were taken from *M. tb* pellicle cultures after 5 weeks of growth and placed on poly-L-lysine-treated plastic coverslips (13 mm, Thermonax plastic for cell culture). Samples were then fixed overnight in a 4% formaldehyde, 2% glutaraldehyde solution in Dulbecco's phosphate-buffered saline (-calcium, -magnesium) (DPBS) (Hyclone Laboratories Inc, Logan, UT). Following fixation, samples were washed with DPBS, then treated with 1% osmium tetroxide for 1 hr. Following osmium tetroxide treatment, samples were washed with DPBS. Next, cells underwent ethanol dehydration, which was followed by critical point drying. Following that, samples were then placed on aluminum stubs and sputter-coated with 20 nm platinum. The samples were imaged at 3 kV by a Zeiss GeminiSEM 450 SEM.

To measure cell length via SEM, SEM images taken at 10,000× were opened on ImageJ. The pixel length of the scale bar was measured with the 'straight' tool. Next, the image was cropped to remove any area without *M. tb* cells. Six frames per image were selected using the 'unbiased frames' macro in ImageJ. The cell length in pixels of a fully exposed *M. tb* cells was measured in each frame using the 'straight' tool. A random number generator was used to select frames for the remaining four cell lengths. After all cell lengths were measured, pixel length was converted to µm using the scale bar prior to analysis. Cell lengths were measured across two biological replicates (two independent cultures) for all strains except SK540-0, where only one replicate was available.

## Planktonic growth curves

To compare rates of growth between ancestral and evolved populations, we inoculated 5 mL 7H9OTG with 250 µL thawed culture from a previously frozen cryovial for each of the ancestral strains and passage 8 of each of the pellicle evolved strains. Cultures were incubated at 37°C with shaking, until grown to an $OD_{600}$~1. 15 mL fresh 7H9OTG was seeded at a starting $OD_{600}$ of 0.02 in triplicate and incubated at 37°C with shaking. $OD_{600}$ measurements were taken every day for 10 days (starting at day 2).

## DNA extraction

gDNA for whole genome sequencing was isolated by following a modified Qiagen DNeasy Blood and Tissue Kit protocol. Briefly, cultures were pelleted and resuspended in Tris-EDTA (TE) buffer in 2 mL screw-cap tubes containing ~250 µg of 0.1 mm glass beads. The samples were heated to 80°C for 50 min to sterilize the suspensions, then frozen at –80°C for at least 1 hr to aid cellular lysis. Tubes were vortexed for 3 min for mechanical lysis. Suspensions were then incubated with 3% w/v mutanolysin (Sigma-Aldrich, M9901) for 1 hr at 37°C. Beads and cellular debris were pelleted, and supernatants were transferred to tubes for DNA purification following the protocol detailed in the Qiagen DNeasy Blood and Tissue Handbook, starting with step 4 of 'protocol: pretreatment for gram-positive bacteria'.

## Library preparation and sequencing

Library preparation was performed using a modified Nextera protocol as described by *Baym et al., 2015* with a reconditioning PCR with fresh primers and polymerase for an additional five PCR cycles to minimize chimeras and a two-step bead-based size selection with target fragment size of 650 bp. Libraries were submitted to the University of Wisconsin-Madison Biotechnology Center (UWBC). Quality and quantity of the finished libraries were assessed using an Agilent DNA High Sensitivity chip (Agilent Technologies, Santa Clara, CA) and Qubit dsDNA HS Assay Kit, respectively. Libraries were standardized to 2 µM. Samples were sequenced generating paired end reads using Illumina HiSeq 2000. The majority of samples (87/101) were sequenced using the above protocol; however, for a

small number of samples that needed resequencing, gDNA was submitted to the UWBC and to the Microbial Genome Sequencing Center for both library prep and sequencing using Illumina NovaSeq 6000 and Illumina NextSeq 2000, respectively.

## Reference-guided assembly and variant calling

Genome assemblies were performed using an in-house reference-guided assembly pipeline (*Youngblom, 2021*). Briefly, raw data was checked for quality with FastQC v0.11.8 (*Andrews, 2018*) and trimmed using TrimGalore v0.6.4 (*Krueger, 2019*). Reads were mapped to the H37Rv reference genome using BWA-MEM v0.7.12 (*Li, 2013*), and Samtools v1.11 (*Li et al., 2009*) view and sort were used to process SAM and BAM files. Picard v1.183 (*Broad Institute, 2022*) was used to remove duplicates and add read information. Reads were locally realigned using GATK v3.5 (*DePristo et al., 2011*) and variants were identified using Pilon v1.16 (*Walker et al., 2014*). Finally, assembly quality was determined using Qualimap BamQC (*Okonechnikov et al., 2016*).

## Lineage typing and phylogenetic tree

We used SNP-IT (*Lipworth et al., 2019*) to identify the lineages of our ancestral strains. To identify the sub-lineage of our ancestral strains, we obtained an alignment of known L4 isolates from a collaborator. We created an SNP alignment of the assemblies from the RGAPepPipe using Snp-sites v 2.4.1 (*Page et al., 2016*) and constructed a maximum likelihood phylogeny using RAxML v8.2.3 (*Stamatakis, 2014*). Next, we visualized the phylogeny using Dendroscope (*Huson and Scornavacca, 2012*) and lineage typed our isolates by comparing their location in the phylogeny to known L4 subtypes. The phylogenetic tree in *Figure 1* was plotted and colored in R using ggtree (*Yu et al., 2017*).

## Identification of fixed SNPs

To identify genetic changes occurring after repeated passaging as a pellicle or planktonic culture, we used Popoolation2 v1.201 (*Kofler et al., 2011b*) and breseq v0.35.0 (*Deatherage and Barrick, 2014*). BAM files from RGAPepPipe (see above) were filtered for quality and used with Samtools mpileup (*Li et al., 2009*) to tabulate mutation frequencies across sequenced timepoints. The mpileup file was converted to a sync file using Popoolation2 and finally converted the sync file to a format suitable for downstream analysis using Python (code available at https://github.com/myoungblom/mtb_ExpEvo, *Smith, 2022* copy archived at swh:1:rev:5ed59742dc9219e32924c7450c77dc03b3663232). Variants were filtered for quality by minimum coverage of 20×, minimum alternate allele count of 5, and a minimum allele frequency of 5%. We also removed variants in repetitive families of genes (PE, PPE, and PE-PGRS) from analysis. Sequence data were also assembled and variants called and annotated using breseq v0.35.0 (*Deatherage and Barrick, 2014*) to confirm variants called using Popoolation2. Breseq was run in population mode for each sequenced timepoint, with all default parameters except one: 'polymorphism-minimum-total-coverage-each-strand' was set to 4. Then the mutational frequencies of each timepoint were compared using gdtools (*Deatherage and Barrick, 2014*) compare run with default parameters. To identify mutations that had significant frequency changes throughout the experiment, we looked for variants that were at a low starting frequency and rose to fixation (>95% frequency) or conversely, started at intermediate frequency and disappeared from the population (<5% frequency).

## Structural variant detection

To identify potential insertions, deletions, and duplications of interest in our genomic sequence data, we called structural variants using Pilon v1.16 as a part of RGAPepPipe. We removed all insertions, deletions, and duplications that were located entirely in regions annotated as repetitive families of genes (PE, PPE, and PE-PGRS). We also removed variants that were identified in ancestral strains and/or that did not remain fixed in subsequent passages. Additionally, we manually curated potential variants by visually inspecting the alignments in Integrative Genomics Viewer (IGV) (*Robinson et al., 2011*; *Thorvaldsdottir et al., 2012*) and removed variants that appeared to be mis-mappings or in regions of poor coverage. Sliding window coverage plots to visualize duplications were made using Samtools bedcov (*Li et al., 2009*) using in-house scripts (code available at https://github.com/myoungblom/mtb_ExpEvo, *Smith, 2022* copy archived at swh:1:rev:5ed59742dc9219e32924c7450c77dc03b3663232). Relative coverage was calculated by dividing each window coverage by the

average coverage across the assembly – as given by BamQC (*Okonechnikov et al., 2016*) – and plotted in R.

## GO term enrichment analysis

To identify possible functional enrichments in the genes within the large duplication that evolved in MT31 and MT55 pellicles (*Figure 5*), we performed a GO term enrichment analysis. GO terms for all *M. tb* genes were downloaded from Quick GO (https://www.ebi.ac.uk/QuickGO/). We used the R package topGO (*Alexa and Rahnenfuhrer, 2022*) to perform an enrichment analysis of the genes present in one or both duplications (a total of 172 genes). We calculated significance using a Fisher's exact test and filtering for p<0.01 (*Supplementary file 3*). We performed the same enrichment analysis on 100 randomly selected, contiguous sets of 172 genes to test the greater significance of our results (*Figure 8—figure supplement 2*).

## Genotype plots

To better understand the structure of the population over passaging, we created genotypes by clustering variants with similar trajectories throughout the experiment. Output from Popoolation2, as described above, was filtered for based on ≥30% frequency change between the final passage and ancestral populations, as well as filtering out all mutations within 1000 bp of each other (we identified that these mutations are often the result of repetitive regions and/or poor mapping). Lolipop (*cdeitrick and pepepdodiu, 2020*) was used with default parameters to cluster mutations into genotypes and create plots.

## Population genetics statistics

Population genetics estimates (Tajima's D, pi, and theta) were calculated as previously described (*O'Neill et al., 2015*). Briefly, we used Popoolation (*Kofler et al., 2011a*) to estimate these parameters in sliding windows across the genome. To reduce biases caused by variable coverage, we randomly sub-sampled read data without replacement to uniform coverage of 50×; we repeated this process nine times to further reduce biases. Genome-wide averages were calculated by averaging all windows across the genome, across all replicate sub-sampled data. Scripts available at https://github.com/myoungblom/mtb_ExpEvo, (*Smith, 2022* copy archived at swh:1:rev:5ed59742dc9219e32924c7450c77dc03b3663232).

## Network analysis

GOI were defined as coding genes containing or just downstream of the fixed mutations, we identified in populations passaged as pellicles (*Figure 5*). Network analysis of GOI was performed by accessing the TFOE data available in the MTB Network Portal (http://networks.systemsbiology.net/mtb/content/TFOE-Searchable-Data-File). TFs identified as significantly contributing to regulation of more than one of our GOIs were included in *Figure 9*. We also used this same dataset to identify GOIs which are themselves regulators and affect expression of other GOIs.

## Fixed SNPs in global datasets

We searched for the presence of mutations identified in our study in publicly available sequencing data using a searchable Compact bit-sliced signature (COBS) index of bacterial genomes curated from the European Nucleotide Archive (*Blackwell et al., 2021*). We used the Python interface to search the COBS index (*Bingmann et al., 2019*). Our query sequences included 50 bp in either direction of the SNP of interest, with a k-mer matching threshold of 1. To determine the frequency of isolates in the COBS index with each SNP of interest, we needed to determine the number of *M. tb* genomes in the database: we searched the database using the full H37Rv 16S rRNA sequence (accession: NR_102810.2) which returned 39,941 results – this total number of *M. tb* isolates was used to calculate frequency of our SNPs of interest (*Figure 5*).

## RNA isolation

Frozen cryovials of ancestral populations (MT540-0, MT49-0) and evolved populations (MT540-4, MT49-12) were thawed, and 250 µL of each was inoculated into 25 mL 7H9OTG and incubated at 37°C with shaking and grown to an $OD_{600}$ of ~1. 250 µL of culture was used to inoculate 25 mL Sauton's for biofilm growth. The remaining culture was pelleted, resuspended in 3 mL RNAprotect Bacteria

Reagent (Qiagen), divided into 3×1 mL aliquots (three technical replicates), and frozen at –80°C until RNA extraction. Biofilm cultures were grown as previously described, and pellicles were harvested at 5 weeks. Pellicles were broken up and suspended in 3 mL RNAprotect, divided into 3×1 mL aliquots (three technical replicates), and frozen at –80°C until RNA extraction. RNA was extracted from each 1 mL aliquot (three technical replicates per sample), using the illustra RNAspin Mini RNA Isolation Kit (GE Healthcare) following the 'Protocol for total RNA purification from up to $10^9$ bacterial cells,' which includes a 15 min room temperature incubation with RNase-free DNase I. RNA integrity was assessed by bleach gel electrophoresis (*Aranda et al., 2012*; ) and quantified using a NanoDrop ND-1000 spectrophotometer (NanoDrop Technologies Inc).

## Real-time quantitative qPCR

Approximately 100 ng of each RNA sample was used for cDNA synthesis with random hexamer primers using the RevertAid First Strand cDNA Synthesis Kit (Thermo Scientific) and included a 5 min incubation at 65°C and a reaction temperature of 44°C for GC-rich templates. Real-time quantitative PCR was performed in EU Fast Cycler PCR 96-well plates (BIOplastics) using the StepOnePlus Real-Time PCR System (v. 2.3, Applied Biosystems). We used sets of primers previously published and verified (*Akhtar et al., 2009*) to measure the expression of *lpdA* normalized to the expression of the endogenous control, *sigA*. All real-time PCR assays were run in a total reaction volume of 20 µL comprised 2× Fast SYBR Green Master Mix (Applied Biosystems), 200 nM of both forward and reverse primers (Integrated DNA Technologies, Coralville, IA, USA), and 3 µL of cDNA. RT-qPCR cycling parameters were set as follows: an initial AmpliTaq Fast DNA Polymerase, UP Activation step of 20 s at 95°C, followed by 40 cycles of 3 s at 95°C and 30 s at 60°C. Each reaction was repeated three times with three independent cDNA samples. Negative controls consisting of no-template reaction mixtures were run with all reactions. Melting-curve analysis was carried out to confirm the specificity of the amplified product. After baseline correction and determination of the threshold settings, the Factor-qPCR ratio method (*Ruijter et al., 2015*) was used to remove between-run variation due to the experiment including multiple plates. The factor corrected ΔCT values of the three technical replicates (MT540-EB has only 2 replicates) were then averaged, and the ΔΔCTs were calculated by subtracting a reference value (MT540-AP) from each sample. The expression fold change was calculated using the $2^{-\Delta\Delta CT}$ method (*Livak and Schmittgen, 2001*), as PCR efficiencies had previously been found to be similar. Low and high fold changes were calculated by adding or subtracting the SD from the fold change. The results are expressed as log2 of the fold changes. Kruskal–Wallis test was used to determine the presence of significant log2 fold differences in the dataset (p-value=1.89e-04). Then, a Wilcoxon test was performed to compare log2 fold changes between sample types (AP, AB, EP, and EB) to identify significant differences between pairs. A p-value <0.05 was considered significant.

## *lpdA* overexpression mutant

The *lpdA* gene was overexpressed in *M. tb* H37Rv at the L5 site using the integrative, constitutive overexpression plasmid pCT94 provided by Sarah Fortune's lab (Harvard T.H. Chan School of Public Health). *lpdA* was amplified and modified to include *NdeI* and *HindIII* digestion sites using forward primer  CCGCATGCTTAATTAAGAAGGGAGATATACA-CCGCCGTCTAAGCTTTCCCCC  and  reverse primer  GACCTCTAGGGTCCCCAATTAATTAGCTAA-TAGCCTGGGCGCTCGCGATAA. After amplification with Platinum SuperFi Green PCR Master Mix (Invitrogen, Waltham, MA), the PCR product was digested with *NdeI* and *HindIII* (NEB, Ipswich, MA). The digested product was then ligated with a *NdeI* and *HindIII* cut pCT94 using Gibson assembly (NEB, Ipswich, MA). The assembly product (CT94-*lpdA*) was then heat-shock transformed into DH5α *Escherichia coli* cells and selected for on LB plates with 50 µg/mL kanamycin overnight. Plasmid integration was verified via *NdeI* and *HindIII* digestion. CT94-*lpdA* and an empty vector control (CT94) were electroporated into electrocompetent *M. tb* H37Rv, and transformants were recovered in 2 mL of fresh 7H9 for 24–48 hr at 37°C, and then struck on 7H10 plates containing 25 µg/mL kanamycin. Both control and *lpdA* overexpression strains were grown as pellicles in triplicate and wet weight measurements taken as described above.

## Acknowledgements

We would like to thank the Fortune lab with their assistance in the design of our overexpression experiments. We would like to acknowledge Soleil Young for her help phenotyping pellicles. We also thank

the University of Wisconsin Biotechnology Center and the Microbial Genome Sequencing Center for providing sequencing services.

# Additional information

## Competing interests

Tracy M Smith: The authors declare no conflict of interes. The other author declares that no competing interests exist.

## Funding

| Funder | Grant reference number | Author |
|---|---|---|
| National Institute of Allergy and Infectious Diseases | R01AI113287 | Caitlin S Pepperell |
| National Science Foundation | DGE-1747503 | Madison A Youngblom |
| National Institute of Allergy and Infectious Diseases | T32-AI55391 | Lindsey L Bohr |
| National Science Foundation | DGE-1256259 | Tatum D Mortimer Mary B O'Neill |
| National Institute of Allergy and Infectious Diseases | T32-GM07215 | Tatum D Mortimer |

The funders had no role in study design, data collection and interpretation, or the decision to submit the work for publication.

## Author contributions

Tracy M Smith, Conceptualization, Formal analysis, Investigation, Methodology, Writing – original draft, Writing – review and editing; Madison A Youngblom, Formal analysis, Investigation, Methodology, Visualization, Writing – original draft, Writing – review and editing; John F Kernien, Investigation, Visualization, Writing – review and editing; Mohamed A Mohamed, Mary B O'Neill, Formal analysis, Writing – review and editing; Sydney S Fry, Lindsey L Bohr, Tatum D Mortimer, Investigation, Writing – review and editing; Caitlin S Pepperell, Conceptualization, Funding acquisition, Methodology, Project administration, Supervision, Writing – original draft, Writing – review and editing

## Author ORCIDs

Madison A Youngblom (ID) http://orcid.org/0000-0003-4816-1113
Mohamed A Mohamed (ID) http://orcid.org/0000-0003-4222-5112
Tatum D Mortimer (ID) http://orcid.org/0000-0001-6255-690X
Caitlin S Pepperell (ID) http://orcid.org/0000-0002-6324-1333

## Ethics

Archived strains of Mycobacterium tuberculosis originally isolated from clinical specimens were analyzed in this study. Research on these strains was reviewed and approved by the University of Wisconsin Madison Institutional Review Board.

## Decision letter and Author response

Decision letter https://doi.org/10.7554/eLife.78454.sa1
Author response https://doi.org/10.7554/eLife.78454.sa2

# Additional files

## Supplementary files

• Supplementary file 1. Outline of data collection for each strain at each passage point.

• Supplementary file 2. Variants with a ≥30% frequency change over the course of four planktonic passages. Grouped by mutations within the same gene and sorted by genes that acquired the most

independent SNPs across replicates and strains.

• Supplementary file 3. Significantly (Fisher's-exact, p<0.01) enriched GO terms within the large duplication that arose in L4.9 strains MT31 and MT55. Of note is enrichment of nucleotide biosynthesis/metabolic processes.

• Supplementary file 4. Mutations present in ancestral pairs of strains which went on to develop convergent mutations in response to biofilm growth (MT31/MT55 and MT49/MT540). 'Ref' and 'alt' refer to the alleles present in the reference strain (H37Rv) and the clinical isolate, respectively. Mutations without an annotation exist in intergenic regions.

• MDAR checklist

### Data availability

Raw sequence data for ancestral and evolved isolates has been submitted to NCBI under the project accession PRJNA720149.

The following dataset was generated:

| Author(s) | Year | Dataset title | Dataset URL | Database and Identifier |
|---|---|---|---|---|
| Smith TM, Youngblom MA | 2022 | Experimental Evolution of Mycobacterium tuberculosis biofilms | https://www.ncbi.nlm. nih.gov/bioproject/ PRJNA720149 | NCBI BioProject, PRJNA720149 |

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
