## [Editor Report]

This study is of relevance to microbiologists interested in understanding the mechanism of biofilm development and notably in *Mycobacterium tuberculosis* (Mtb). The authors describe the use of experimental evolution for identifying genes/loci involved in controlling biofilm formation and report a few fixed mutations/genetic duplications that could be associated with biofilm formation or its regulation. The findings also reveal epistatic interactions among genes that are consequential during growth of Mtb in biofilms.

---

## [Decision Letter]

**Decision letter after peer review:**

[Editors’ note: the authors submitted for reconsideration following the decision after peer review. What follows is the decision letter after the first round of review.]

Thank you for submitting the paper "Rapid adaptation of a complex trait during experimental evolution of *Mycobacterium tuberculosis*" for consideration by *eLife*. Your article has been reviewed by 2 peer reviewers, and the evaluation has been overseen by a Reviewing Editor and a Senior Editor. The following individual involved in review of your submission has agreed to reveal their identity: Anil Ojha (Reviewer #1).

The reviewers have discussed their reviews with one another and we had a joint consultation.

The reviewers are clearly explained their concerns so I will not reiterate them all here except saying that the are 5 key points that deserve your attention and should be addressed.

1. Rationale for strain selection should be provided.

2. Inclusion of additional controls will help interpret the mutations as evolution under selection pressure.

3. Epistasis: sequences analysis of all six genomes, and pairing the pre-existing sequence variations with the evolving mutations.

4. Clarify biofilm vs pathogenesis.

5. Experimental demonstration that an allele leads to increased biofilm formation to support the authors claims.

*Reviewer #1:*

This paper by Smith et al. tests a hypothesis that selection pressure drives evolution of *Mycobacterium tuberculosis* (Mtb) towards biofilm development. The study involves phenotypic and genotypic analyses of six clinical isolates of Mtb upon multiple passages through biofilm growth. The study finds progressive emergence of mutations that eventually get fixed in the subsequent passages. Concomitant with the appearance of these mutations, the strains generally acquired greater propensity to form biofilms, implying that the mutations are likely to confer growth advantage during biofilm development. Moreover, the loci acquiring mutations appear to be distinct for each lineage, but somewhat overlap across different strains within a lineage. The authors identify mutations in two specific loci – lpdA and a 120Kb gene cluster with 172 genes – that appear to be convergent. By further showing that LpdA expression is upregulated in biofilms and that identified mutations increase its expression, the authors suggest an important role for this gene in biofilm formation. In addition, the authors identified mutations associated with the expression of multiple transcription regulators, some of which (e.g. RegX3) have been previously implicated in biofilms. Overall, this is an exciting study that offers new insight into genome optimization necessary for Mtb adaptation to biofilm growth. However, the paper in its current form is incomplete and additional experiments and analyses are necessary for clarity. Specifically, the manuscript needs revision in the following areas:

– A clear rationale for strain selection should be provided.

– Inclusion of additional controls will help interpret the mutations as evolution under selection pressure.

– Epistatic interactions among genes should be further analyzed in the context of pre-existing genetic variations among the strains. This will likely provide a more specific nature of allele combinations promoting Mtb growth in biofilms.

The crucial gaps listed below should be addressed.

1. Basis of selecting the six strains: The authors should include a clear rationale for selecting these specific six strains. It seems all the six strains have some defect in biofilm formation to start with, raising a question about evolution in the clinical strains that do form biofilms in the first passage. Doesn't this approach raise bias in the study?

2. How does experimental evolution differ from isolation of suppressor mutation of a biofilm defective mutant? This distinction should be discussed.

3. Some key controls in the experiments are missing. A necessary control would be the genomic analysis of the strains in planktonic culture passaged for similar period (at least 4 passages when the mutations in many strains are fixed). How will the trajectory of evolution appear in the absence of the selection pressure? Another important control could be a standard laboratory strain that is able to form biofilm (e.g. H37Rv). Inclusion of both these controls will offer insight into evolution without any selection pressure (the lack of pressure in H37Rv biofilms could be inferred from the fact that the mutations in the strain necessary for biofilm formation are already fixed). These controls therefore could render an important dataset for analyzing evolution under selection pressure of biofilm growth.

4. Epistasis: The lineage-specific convergence of mutations is remarkable and in my opinion the highlight of this study. However, this observation also begs the question as to what allelic combinations are necessary for genome optimization for Mtb growth in biofilms. While this has been touched very briefly in the paper, perhaps a careful sequences analysis of all six genomes, and pairing the pre-existing sequence variations with the evolving mutations could render some new insight.

5. Biofilm vs pathogenesis: In line 281-288, the authors report the rare nature of the evolved mutations in biofilms, with an implication that the loci are under "strong purifying selection" in the natural habitat of Mtb. Do the authors mean to suggest that the natural habitat of Mtb (host tissues) is not conducive for biofilm formation? If so, then this would apparently contradict the fact that mutations leading to higher expression of LpdA, a virulence factor, also promote biofilm formation. In fact, published literature also report multiple genes involved in virulence are required for biofilm formation, including those regulated by RegX3. Moreover, biofilm-like aggregates of Mtb have been reported in vivo by many studies, as cited in this paper. I suggest authors should clarify this point further.

*Reviewer #2:*

The manuscript titled "Rapid adaptation of a complex trait during experimental evolution of Mycobacterium" by Smith et al. describes the use of experimental evolution for identifying genes/loci involved in controlling biofilm formation. The authors exposed six related strains from three subclades to the experimental evolution through successive growth as pellicle biofilms. After exposure to experimental pressure, the genomes were sequenced to map the loci associated with the biofilm formation/regulation. The authors then characterized several features such as pellicle architecture, cell length, wet weight of biofilm, growth rate, etc., and correlated them with the passage. The authors also utilized DNA sequencing for identifying genetic changes associated with experimental pressure. A handful of genetic loci were identified through this approach that could be related to biofilm formation and its regulation.

The primary strength of the manuscript is the finding that bacterial cells become geared up for forming thick biofilms with high amounts of EPS upon experimental evolution. This is anecdotal to the observation that lab strain of Mtb H37Rv leads to productive infection upon passaging in animals. This capacity is lost on subsequent culturing in vitro. However, authors have ignored that such phenotypes/morphotypes are reversible and generally regulated at the epigenetic level rather than at the genetic levels that lead to inheritance and permanency of the phenotype. Nevertheless, the authors have identified a few genetic loci that may be associated with stringent biofilm morphotypes. Given that it was claimed that these loci are involved in biofilm formation and or its regulation, experiments wherein the introduction of an allele leads to increased biofilm formation are needed to justify the claim. Without such experiments, these claims are not justified.

Besides, the comments in public review, I have the following specific comments for the authors.

– Data presented in figure 4 A, B, and C will improve by inducing data on H37Rv passages as well. It seems that the same growth curve of H37Rv for the three experiments is used in these panels. This should be clarified in the figure legends.

– Discussion may be shortened to improve the readability of the manuscript. At the same time, I would suggest briefly discussing the role of epigenetic factors in the context of the current study.

– I also feel that changes in the cell length and relative fitness for planktonic growth are not clearly correlated, i.e., some strains become larger, while others were smaller. Thus, authors may improve the readability of the manuscript by emphasizing more on the consistent morphotypes such as the increase in ECM and wet weight of pellicle formed by the passaged cells.

---

## [Author Response]

The reviewers have discussed their reviews with one another and we had a joint consultation.The reviewers are clearly explained their concerns so I will not reiterate them all here except saying that the are 5 key points that deserve your attention and should be addressed.1. Rationale for strain selection should be provided.

See response to Reviewer #1, point #1.

2. Inclusion of additional controls will help interpret the mutations as evolution under selection pressure.

See response to Reviewer #1, point #3.

3. Epistasis: sequences analysis of all six genomes, and pairing the pre-existing sequence variations with the evolving mutations.

See response to Reviewer #1, point #4.

4. Clarify biofilm vs pathogenesis.

See response to Reviewer #1, point #5.

5. Experimental demonstration that an allele leads to increased biofilm formation to support the authors claims.

See response to Reviewer #2, point #2.

Reviewer #1:This paper by Smith et al. tests a hypothesis that selection pressure drives evolution of Mycobacterium tuberculosis (Mtb) towards biofilm development. The study involves phenotypic and genotypic analyses of six clinical isolates of Mtb upon multiple passages through biofilm growth. The study finds progressive emergence of mutations that eventually get fixed in the subsequent passages. Concomitant with the appearance of these mutations, the strains generally acquired greater propensity to form biofilms, implying that the mutations are likely to confer growth advantage during biofilm development. Moreover, the loci acquiring mutations appear to be distinct for each lineage, but somewhat overlap across different strains within a lineage. The authors identify mutations in two specific loci – lpdA and a 120Kb gene cluster with 172 genes – that appear to be convergent. By further showing that LpdA expression is upregulated in biofilms and that identified mutations increase its expression, the authors suggest an important role for this gene in biofilm formation. In addition, the authors identified mutations associated with the expression of multiple transcription regulators, some of which (e.g. RegX3) have been previously implicated in biofilms. Overall, this is an exciting study that offers new insight into genome optimization necessary for Mtb adaptation to biofilm growth. However, the paper in its current form is incomplete and additional experiments and analyses are necessary for clarity. Specifically, the manuscript needs revision in the following areas:– A clear rationale for strain selection should be provided.– Inclusion of additional controls will help interpret the mutations as evolution under selection pressure.– Epistatic interactions among genes should be further analyzed in the context of pre-existing genetic variations among the strains. This will likely provide a more specific nature of allele combinations promoting Mtb growth in biofilms.The crucial gaps listed below should be addressed.1. Basis of selecting the six strains: The authors should include a clear rationale for selecting these specific six strains. It seems all the six strains have some defect in biofilm formation to start with, raising a question about evolution in the clinical strains that do form biofilms in the first passage. Doesn't this approach raise bias in the study?

Strains were chosen from our collection of ~500 clinical isolates based on genetic background and biofilm phenotype. We chose isolates that are relatively closely related but that exhibited different biofilm phenotypes prior to passaging in vitro. In this way, we could assess the impact of both genetic and phenotypic backgrounds on subsequent adaptation. We used a nested design with regards to phylogenetic structure of the study sample, with pairs of strains from each clade and the inclusion of two closely related clades (L4.4.1.1 and L4.1.1.2) and a third slightly more distantly related clade (L4.9). Prior to this study, it was known that distantly related strains exhibited different biofilm phenotypes (Pang et al. 2012); we add to the literature here an observation that subtle differences in genetic background can affect not only biofilm phenotype but also adaptation to a new environment.

In our large collection of clinical isolates, we have observed a continuum of biofilm phenotypes including isolates that do not form a visible biofilm. None of these isolates can be considered defective in the sense that they were all successful in establishing infection, and our observations indicate that biofilm production is non-binary, with biofilm growth exhibiting a spectrum of phenotypes. Our rationale for selecting the strains included in this experiment was to choose a sample with a range of starting biofilm phenotypes, with some forming robust biofilms (ie MT345) and others forming no visible biofilm (ie MT72) at baseline. We find that regardless of starting phenotype, all strains evolved more robust biofilms, and even that strains with different ancestral phenotypes evolved similar mutations (MT31 and MT55 duplication). This has been clarified in the “Sample” section of the results.

2. How does experimental evolution differ from isolation of suppressor mutation of a biofilm defective mutant? This distinction should be discussed.

We do not consider any of our strains defective – these strains all successfully established culture-positive infections and were directly isolated from patients. As noted above, they exhibited a range of biofilm phenotypes prior to the imposition of selection during passaging. The goal of our study was to direct further evolution of these strains to form more robust biofilms to identify genes and pathways involved in biofilm formation.

3a. Some key controls in the experiments are missing. A necessary control would be the genomic analysis of the strains in planktonic culture passaged for similar period (at least 4 passages when the mutations in many strains are fixed). How will the trajectory of evolution appear in the absence of the selection pressure?

We have added genomic data from passaging our strains as planktonic cultures:

three independently evolving replicates for each strain, four passages for each replicate. We would like to note that passaging as a planktonic culture is not an absence of selective pressure, but rather a different pressure than we induced during pellicle passage. That said, with these new data we are able to assess whether the mutations we identified as associated with more robust biofilm growth are general lab adaptations – i.e., mutations that arose after both planktonic and pellicle passage indicating a fitness advantage common to both environments. Our results show that the mutations in Table 1 are specific to enhanced growth in the pellicle environment. We did identify distinct mutations associated with planktonic passaging, and some similar observations to the pellicle passaging such as an effect of genetic background (epistasis).

Another important control could be a standard laboratory strain that is able to form biofilm (e.g. H37Rv). Inclusion of both these controls will offer insight into evolution without any selection pressure (the lack of pressure in H37Rv biofilms could be inferred from the fact that the mutations in the strain necessary for biofilm formation are already fixed). These controls therefore could render an important dataset for analyzing evolution under selection pressure of biofilm growth.

As mentioned above in point #1, some of our ancestral strains made robust biofilms at baseline – specifically MT345 (Figure 3). Therefore, we have already included this control in our experimental design. Interestingly, we found that even though M345 formed a thick biofilm at baseline, it underwent one of the most dramatic changes in response to selection pressure in our experiment. This suggests that pellicle growth can be enhanced even among strains that are well adapted to this form of growth.

Part of our rationale for choosing clinical isolates for this study was to avoid the use of strains that have been heavily manipulated in the lab. H37Rv is well adapted to the lab environment, specifically to planktonic growth (see Figure 4) as it was not deliberately passaged as a pellicle. We do not believe H37Rv is likely to stop evolving in response to selection for pellicle growth (or other phenotypes). Indeed, while H37Rv strains form robust pellicles, BCG strains generally form heavier pellicles reflecting their years of deliberate passaging as a pellicle. This indicates that there is likely a large adaptive space left for H37Rv pellicle growth, as we observed with clinical isolates.

We also note that unfortunately (or fortunately, depending on your perspective), it is not possible to manipulate bacterial strains in the lab without imposing a selection pressure. The selection pressures vary according to the specific type of manipulation, but bacterial strains will continue to evolve and adapt: this is reflected, for example, in the observation that H37Rv strains from different laboratories are genetically distinct and that virulence factors important for infection (eg. PDIM) can be lost even after minimal in vitro manipulation of bacterial strains (Domenech and Reed 2009; Ioerger et al. 2010). As discussed in the manuscript, experimental evolution of bacteria suggests that early adaptation is characterized initially by broad phenotypic leaps, but incremental changes can continue to occur over extended passaging.

4. Epistasis: The lineage-specific convergence of mutations is remarkable and in my opinion the highlight of this study. However, this observation also begs the question as to what allelic combinations are necessary for genome optimization for Mtb growth in biofilms. While this has been touched very briefly in the paper, perhaps a careful sequences analysis of all six genomes, and pairing the pre-existing sequence variations with the evolving mutations could render some new insight.

We have included these data in the revised manuscript (SupplementaryData_1). Analysis of the strains within genetic backgrounds that adapted convergently reveals 36 mutations specific to L4.9 and 206 mutations specific to L4.4.1.1.

5. Biofilm vs pathogenesis: In line 281-288, the authors report the rare nature of the evolved mutations in biofilms, with an implication that the loci are under "strong purifying selection" in the natural habitat of Mtb. Do the authors mean to suggest that the natural habitat of Mtb (host tissues) is not conducive for biofilm formation? If so, then this would apparently contradict the fact that mutations leading to higher expression of LpdA, a virulence factor, also promote biofilm formation. In fact, published literature also report multiple genes involved in virulence are required for biofilm formation, including those regulated by RegX3. Moreover, biofilm-like aggregates of Mtb have been reported in vivo by many studies, as cited in this paper. I suggest authors should clarify this point further.

We are in agreement with the reviewer about the relevance of biofilms to natural infection and agree that the identification of bacterial aggregates and biofilm biomarkers in human tissues is compelling. Our working hypothesis, based on the existing literature and observations from this study, is that *M. tb* forms biofilms during natural infection but that this trait is tightly regulated and thus biofilm growth occurs only in response to specific stimuli. We hypothesize that the mutations observed in this experiment are not observed in nature because they result in global de-repression of biofilm formation, which could be detrimental during natural infection for many potential reasons such as the energetic costs of matrix production and their immunostimulatory properties. We have also noted overlaps between genes involved in biofilm formation and other aspects of virulence and did not mean to imply that these traits were segregated. We have edited the Discussion to clarify these points.

Reviewer #2:The manuscript titled "Rapid adaptation of a complex trait during experimental evolution of Mycobacterium" by Smith et al. describes the use of experimental evolution for identifying genes/loci involved in controlling biofilm formation. The authors exposed six related strains from three subclades to the experimental evolution through successive growth as pellicle biofilms. After exposure to experimental pressure, the genomes were sequenced to map the loci associated with the biofilm formation/regulation. The authors then characterized several features such as pellicle architecture, cell length, wet weight of biofilm, growth rate, etc., and correlated them with the passage. The authors also utilized DNA sequencing for identifying genetic changes associated with experimental pressure. A handful of genetic loci were identified through this approach that could be related to biofilm formation and its regulation.The primary strength of the manuscript is the finding that bacterial cells become geared up for forming thick biofilms with high amounts of EPS upon experimental evolution. This is anecdotal to the observation that lab strain of Mtb H37Rv leads to productive infection upon passaging in animals. This capacity is lost on subsequent culturing in vitro. However, authors have ignored that such phenotypes/morphotypes are reversible and generally regulated at the epigenetic level rather than at the genetic levels that lead to inheritance and permanency of the phenotype. Nevertheless, the authors have identified a few genetic loci that may be associated with stringent biofilm morphotypes.

While we agree with the reviewer about the importance of epigenetic mechanisms in shaping *M. tb* transcriptional responses, the phenotypic changes we observed in pellicle passaged strains were more stable than we would expect for phenomena arising from transient changes in gene expression. Phenotypic assays (wet weights, Figure 3 and growth curves, Figure 4) were performed following a cycle of planktonic growth, rather than carrying the pellicle passaging forward into these assays. At the conclusion of the experiment, the strains were frozen and subsequently revived in a single planktonic culture prior to performing the phenotypic assays.

We did not observe any mutations at methyltransferase alleles but do not discount the possibility that the mutations we did observe had an effect on methylation or other epigenetic changes by unknown mechanisms. This has been clarified in the results. Although we considered further ‘back-passaging’ as a planktonic culture, this would exert distinct selection pressures, as shown by our planktonically passaged populations, that would interfere with the observations specific to pellicle conditions. We have added a section to the discussion RE epigenetics.

Given that it was claimed that these loci are involved in biofilm formation and or its regulation, experiments wherein the introduction of an allele leads to increased biofilm formation are needed to justify the claim. Without such experiments, these claims are not justified.

We agree with the reviewer that more experimental evidence is needed to clarify the role of *lpdA* in biofilm formation. To address this, we measured pellicle wet weights after the introduction of a second copy of *lpdA* on a constitutive, integrative plasmid. After inserting a second copy of *lpdA* we identified significant increases in biofilm wet weight (Figure 6C) without affecting qualitative biofilm phenotype (Figure S4).

– Data presented in figure 4 A, B, and C will improve by inducing data on H37Rv passages as well. It seems that the same growth curve of H37Rv for the three experiments is used in these panels. This should be clarified in the figure legends.

See response to Reviewer #1, point #3b.

– Discussion may be shortened to improve the readability of the manuscript. At the same time, I would suggest briefly discussing the role of epigenetic factors in the context of the current study.

Discussion has been clarified, some sections have been removed and a section on epigenetics added.

– I also feel that changes in the cell length and relative fitness for planktonic growth are not clearly correlated, i.e., some strains become larger, while others were smaller. Thus, authors may improve the readability of the manuscript by emphasizing more on the consistent morphotypes such as the increase in ECM and wet weight of pellicle formed by the passaged cells.

The results and discussion have been clarified to better emphasize the consistent changes in phenotype observed across all strains.